# Recent Developments in PET and SPECT Radiotracers as Radiopharmaceuticals for Hypoxia Tumors

**DOI:** 10.3390/pharmaceutics15071840

**Published:** 2023-06-27

**Authors:** Anh Thu Nguyen, Hee-Kwon Kim

**Affiliations:** 1Department of Nuclear Medicine, Jeonbuk National University Medical School and Hospital, Jeonju 54907, Republic of Korea; thu.ngnanh39@gmail.com; 2Research Institute of Clinical Medicine of Jeonbuk National University-Biomedical Research Institute of Jeonbuk National University Hospital, Jeonju 54907, Republic of Korea

**Keywords:** hypoxia, tumor, radiopharmaceuticals, PET, SPECT

## Abstract

Hypoxia, a deficiency in the levels of oxygen, is a common feature of most solid tumors and induces many characteristics of cancer. Hypoxia is associated with metastases and strong resistance to radio- and chemotherapy, and can decrease the accuracy of cancer prognosis. Non-invasive imaging methods such as positron emission tomography (PET) and single-photon emission computed tomography (SPECT) using hypoxia-targeting radiopharmaceuticals have been used for the detection and therapy of tumor hypoxia. Nitroimidazoles are bioreducible moieties that can be selectively reduced under hypoxic conditions covalently bind to intracellular macromolecules, and are trapped within hypoxic cells and tissues. Recently, there has been a strong motivation to develop PET and SPECT radiotracers as radiopharmaceuticals containing nitroimidazole moieties for the visualization and treatment of hypoxic tumors. In this review, we summarize the development of some novel PET and SPECT radiotracers as radiopharmaceuticals containing nitroimidazoles, as well as their physicochemical properties, in vitro cellular uptake values, in vivo biodistribution, and PET/SPECT imaging results.

## 1. Introduction

Hypoxia is defined as a state in which supplies of oxygen (O_2_) to tissue are insufficient for biological functions [1,2]. The unceasing growth of cancer cells leads to abnormalities in the structure and function of tumor vessels which decrease supplies to the tumor, especially the tumor interior. The proliferation of cancer cells also increases the glucose metabolism and oxygen consumption of the tumor. The net effect of inadequate supplies and the high consumption of O_2_ causes hypoxia in the tumor [2,3].

Hypoxia is a common feature of most solid tumors. Oncological hypoxia often has oxygen levels below 1000 ppm [4]. Many studies have demonstrated that hypoxia promotes the malignant progression of uteri cervix cancer, prostate cancer, glioblastoma, gastric cancer, and so on [5,6,7,8,9]. Hypoxia-inducible factors (HIFs) are transcription factors that mediate the cellular response to hypoxia via the transcription of various hypoxia-inducible genes. The overexpression of HIFs is usually found in solid tumors. Aside from HIF3, which has an unclear role, HIF1 and HIF2 regulate many tumor survival and growth factors [10]. Via the mechanism involving HIFs, many hallmarks of cancer can be induced by hypoxia; therefore, tumors can adapt, overcome the lack of O_2_, invade, and metastasize [11,12,13,14,15]. Several metastases are associated with hypoxia, for instance, soft tissue sarcoma, breast cancer, gynecological cancer, and pancreatic cancer [16,17,18,19,20,21]. For many years, hypoxia has been directly or indirectly related to therapeutic resistance of radio and chemotherapy [6,22,23,24]; thus, it can adversely affect the prognosis of cancer [9,25]. Many studies have also reported correlations between hypoxia and poor outcomes and patient mortality [5,26,27]. Owing to the relationship between hypoxia and malignant progression, radioresistance, chemoresistance, and treatment failure, tumor hypoxia is considered a helpful prognostic factor; hence, an assessment of hypoxia is necessary for aggressive cancer treatment [22].

Tumor hypoxia can be detected via invasive or non-invasive methods. Several invasive methods have been developed to measure hypoxia. The measurement of O_2_ tension (pO_2_) is carried out by directly inserting an oxygen microelectrode into the tumor [28]. The use of HIF expression as a hypoxia marker has been studied by using binding assays with fluorescent or immunohistochemical antibodies on biopsy specimens [29,30,31]. In addition, a comet assay can directly measure radiation-induced damage to DNA in individual cells [32,33,34]. However, these invasive methods have many drawbacks which limit their applications for detecting hypoxia in patients. In particular, only small samples can be taken from tumors, the tumors have to be easily accessible, and specialized personnel and equipment are also required. Moreover, low reproducibility, the heterogeneous distribution of hypoxia, and sampling mistakes can also affect the effectiveness of invasive methods. These limitations have encouraged the development of non-invasive methods using radiolabeled imaging agents [29,35].

Molecular imaging is defined as the visualization, characterization, and quantification of biological processes in intact living subjects using specific imaging probes [36,37,38,39,40,41,42,43,44,45,46,47,48,49,50,51,52,53,54,55]. Nowadays, molecular imaging technologies provide insights into biological events, pathology, and the mechanisms of diseases; thus, they play an important role in various fields of neuroscience, cardiology, gene therapy, oncology, and so on [56,57,58,59,60,61,62,63,64,65,66,67,68]. To visualize biological events in a non-invasive way, a variety of molecular imaging methods have been developed. These include anatomic imaging methods such as computed X-ray tomography (CT) and magnetic resonance imaging (MRI), nuclear imaging methods, such as positron emission tomography (PET) and single-photon emission computed tomography (SPECT), optical imaging methods, and hybrid imaging (or multimodality imaging) methods such as PET/CT, SPECT/CT, and PET/MR [69,70,71,72,73,74,75,76].

PET and SPECT are non-invasive techniques for visualizing physiological and biological processes by detecting radioisotope-emitting positions, or detecting electron or gamma emissions [77]. Since their first descriptions in 1950 and 1963 [78,79], PET and SPECT are used in numerous biological and clinical applications because of their high sensitivity, deep penetration, and range of radiopharmaceuticals [58]. They have both been effectively used in various studies for the diagnosis and therapy of many diseases [80,81,82,83,84,85,86,87] such as vascular diseases [88,89,90,91,92], heart diseases [93,94], central nervous system diseases, including Alzheimer’s and Parkinson’s diseases [95,96,97,98,99], chronic inflammatory diseases [100], and several types of cancer [101,102,103]. In oncology, PET and SPECT not only detect the location of tumors in the body but also have the sensitivity to monitor events occurring in carcinogenesis including the expression of specific proteins and biological processes. Thus, these imaging techniques play a vital role in the detection of cancer. Conventional anatomic imaging techniques like MRI and CT, in most cases, detect tumors with a diameter of one centimeter or more, which already contain more than a billion cells and might include some metastases (<1 cm) [58]. However, PET and SPECT imaging methods have the capability of detecting tumors as small as one millimeter [104,105]. Thus, the use of PET and SPECT for the detection of cancer allows for better cancer diagnosis and staging [106,107], as well as better monitoring of drug responses, and helps to select the appropriate treatment for each patient.

One of the core factors of PET and SPECT development studies is designing and synthesizing appropriate molecular imaging probes, known as radiopharmaceuticals. PET radiopharmaceuticals have been radiolabeled with positron-emitting radionuclides such as ^18^F, ^11^C, ^124^I, ^68^Ga, and ^64^Cu [108]. Among these PET radiopharmaceuticals, ^18^F-labeled radiopharmaceuticals are the most common due to their appropriate physical and chemical properties, and similar steric parameters with hydrogen and low positron energy (0.635 MeV). Importantly, the ^18^F radionuclide has a half-life of 110 min which enables more complicated radiosynthesis, longer in vivo studies, and the distribution to “satellite” PET centers lacking adequate radiochemistry facilities [36]. For instance, [^18^F]FDG, a radiopharmaceutical analog of glucose first used for PET imaging in 1979, has become the most common radiopharmaceutical for clinical studies of cancer [109,110,111]. SPECT radiopharmaceuticals have been radiolabeled with gamma ray emitters such as ^99m^Tc (half-life = 6 h), ^123^I (half-life = 13.3 h), and ^201^Tl (half-life = 73 h) [56]. SPECT radionuclides emit lower-energy photons thus causing lower radiation exposure compared to positron-emitting nuclides; in addition, photons are more scattered and easily absorbed, resulting in a lower photon count, lower sensitivity, and higher image noise in comparison to PET [112].

Nitroimidazole has been an important pharmacophore for the development of radiopharmaceuticals for the detection of hypoxia [113,114]. Nitroimidazole is a class of compounds that includes 2-nitroimidazole, 4-nitroimidazole, 5-nitroimidazole, and metronidazole (or 2-methyl-5-nitroimidazole) (Figure 1). These moieties are considered bioreducible and their mechanism of retention depends on the O_2_ levels within tissues. When entering a viable cell via diffusion, nitroimidazoles are selectively reduced by reductases into potentially reactive nitro radical anions in a process called the activation reaction.

In normal oxygen levels (or under normoxic conditions), this activation reaction is reversible and the nitro radical anions can be immediately reoxidized into the parent nonradical compounds. However, the reaction does not occur in hypoxic conditions. The rate of a reoxidizing reaction depends on the oxygen concentration [114]. Therefore, with the low intracellular oxygen concentration of hypoxic conditions, the reduction of O_2_ and the reoxidization of nitro radical anions cannot compete with the generation of nitro radical anions (Figure 2). Under hypoxia, further reductions of nitro radical anions take place to eventually form reactive species which covalently bind to cellular proteins or DNAs and are trapped within hypoxic cells (Figure 2) [35,113,114,115]. Therefore, hypoxic tissues can be differentiated from normoxic tissues.

Compounds bearing a nitroimidazole moiety have been widely used to detect hypoxia events in bodies. In particular, radiolabeled nitroimidazole compounds have been used for PET or SPECT imaging studies for hypoxia because the accumulation of these radiolabeled compounds in specific locations can be visualized by PET or SPECT. Thus, considerable efforts have been made to develop radiolabeled compounds (radiopharmaceuticals) containing nitroimidazole moieties for preclinical and clinical studies. Several imaging agents such as [^18^F]FMISO [116,117,118], [^18^F]FAZA [119], [^18^F]EF5 [120,121], and [^125^I]IAZA [122] (Figure 3) have been widely developed to study hypoxia; however, these radiotracers still have several drawbacks. An ideal hypoxia radiotracer should exhibit several physicochemical and biological properties. For example, radiotracers for hypoxia should have a high selectivity toward hypoxia with a low retention in normal tissues and a high retention in tumor sites. They should be non-toxic, easy to prepare, convenient [35], and have probable lipophilicity. Aside from hypoxic tissues, the degradation of radiotracers in normal tissues should only generate non-specific metabolites which cannot be trapped in these tissues [35]. Moreover, the trade-off between the absolute tumor uptake signal and the relative tumor/background ratio is also a concern [29]. Thus, various novel radiotracers have been developed to improve the effectiveness of existing radiopharmaceuticals, particularly for pharmacokinetics and biodistribution.

In this review, the development of radiopharmaceuticals radiolabeled with various radioisotopes for PET/SPECT studies of tumor hypoxia between 2014 and the beginning of 2023 have been summarized. In particular, we describe novel radiopharmaceuticals, their physicochemical properties, in vitro biological results, in vivo biodistribution, and PET/SPECT imaging results.

## 2. Development of Radiotracers for Hypoxia

### 2.1. ^18^F Radiotracers for Hypoxia

^18^F is a positron-emitting radioisotope with a half-life of 110 min. Up to now, ^18^F is still the most widely used radioisotope for the preparation of hypoxia-targeting radiopharmaceuticals due to its proper half-life allowing for the extension of PET scans and distribution to distant facilities, as well as low positron energy (0.635 MeV), high electron intensity and high resolution [123,124]. Moreover, ^18^F is small in size and chemically inert, allowing it to easily incorporate into the structures of radiotracers without greatly affecting the physicochemical and biological properties [125,126]. However, the production of ^18^F requires a cyclotron which is high-cost and takes up a large space [127]. Since the development of [^18^F]FMISO, the first radiotracer for imaging of hypoxia, various ^18^F-labeled analogues of 2-nitroimidazole for hypoxia have been extensively studied both preclinically and clinically [128,129].

#### 2.1.1. ^18^F Radiotracers with Linkers for Hypoxia

In 2017, Qiao and co-workers prepared ten ^18^F-labeled polyethylene glycol (PEG)-modified nitroimidazole derivatives [^18^F]**1a**–**c**, [^18^F]**2a**–**c**, and [^18^F]**3a**–**d** (Figure 4) by using click reactions [130]. Ten PEG-modified compounds showed good stability in saline and human serum. The ten compounds were hydrophilic and had log*p* values (ranging from −1.25 ± 0.01 to −0.16 ± 0.01) lower than zero and lower than that of [^18^F]FMISO (0.38 ± 0.08). In vivo biodistribution studies of ten PEG-modified compounds in BALB/c mice bearing EMT-6 tumors suggested that [^18^F]**1a** showed the highest tumor uptake among the ten radiotracers (2.99 ± 0.40%ID/g 1 h p.i.). Although the tumor uptake of [^18^F]**1a** was lower than [^18^F]FMISO (5.43 ± 0.62%ID/g 1 h p.i.), [^18^F]**1a** exhibited higher tumor/liver and tumor/muscle ratios than [^18^F]FMISO. Long chains in the structure of the ten radiotracers partly improved the clearance; however, a large structure might affect the hypoxia-targeting function of nitroimidazole. An in vitro cellular uptake study of radiotracers [^18^F]**1a**–**c** in MCF-7 cells under hypoxic and normoxic conditions indicated that these three compounds showed no significant difference in cellular uptake under hypoxic and aerobic conditions; thus, [^18^F]**1a**–**c** had less selectivity toward hypoxia cells compared to [^18^F]FMISO.

Lin and co-workers developed ^18^F-labeled zwitterion-based ammoniomethyl-trifluoroborate bearing 2-nitroimidazole (^18^F-AmBF_3_-Bu-2NI, [^18^F]**4**) for imaging tumor hypoxia (Figure 5) [131]. [^18^F]**4** had a log*p* value of −1.52 ± 0.02 and remained intact in mouse plasma for 1 h. At 1 h p.i., the in vivo biodistribution of [^18^F]**4** in mice with HT-29 tumors demonstrated minimal uptake in the tumor (0.54 ± 0.13%ID/g), leading to low ratios of tumor/muscle and tumor/blood ratios (0.51 ± 0.25 and 0.99 ± 0.32, respectively). At 3 h p.i., the tumor uptake was reduced to 0.19 ± 0.04%ID/g, whereas tumor/muscle and tumor/blood ratios increased (0.92 ± 0.08 and 2.62 ± 1.02, respectively). However, [^18^F]**4** exhibited lower tumor uptake and tumor/muscle and tumor/blood ratios at 3 h p.i. than [^18^F]FMISO (tumor uptake = 1.84 ± 0.52%ID/g, T/M = 4.52 ± 1.36, T/B = 5.05 ± 0.50). A PET study of [^18^F]**4** in mice bearing HT-29 tumors also showed observable tumors at 3 h p.i. The fast clearance of [^18^F]**4** in normal tissues and organs was consistent with the biodistribution results. An in vitro cellular uptake study using HT-29 cells suggested that [^18^F]**4** was not capable of binding or diffusing across the cell membrane; thus, it could not target nitroreductase inside hypoxic cancer cells.

In 2021, Sun and co-workers prepared nitroimidazole derivative 2-[4-(carboxymethyl)-7-[2-(2-(2-nitro-1H-imidazol-1-yl)acetamido)ethyl]-1,4,7-triazanonan-1-yl]acetic acid (NOTA-NI) as a precursor and radiolabeled it with Al^18^F to produce radiotracer Al^18^F-NOTA-NI ([^18^F]**6**) (Figure 6) [132]. In an in vitro stability test, [^18^F]**6** was stable in human serum for 6 h. Cellular uptake studies in vitro using esophageal cancer cell line ECA109 showed that [^18^F]**6** exhibited better selectivity for hypoxia compared to [^18^F]FMISO. In particular, the hypoxic/normoxic uptake ratio of [^18^F]**6** was 1.53-fold higher than [^18^F]FMISO. The biodistribution of [^18^F]**6** in mice bearing ECA109 xenografts suggested that free [^18^F]**6** tracer was rapidly washed out via the kidneys while [^18^F]FMISO was excreted mostly via the enterohepatic pathway. In comparison to the tumor uptake of [^18^F]FMISO (tumor uptake = 4.45 ± 0.56%ID/g; log*p* = −0.353 ± 0.016), [^18^F]**6** had lower initial tumor uptake (3.61 ± 0.22%ID/g) due to higher hydrophilicity (log*p* = −0.952 ± 0.034) [^18^F]**6** and [^18^F]FMISO exhibited no difference in tumor uptake at 2 h p.i.; however, [^18^F]**6** could provide better contrast with a significantly higher tumor/muscle ratio (2.67 ± 0.08) than [^18^F]FMISO (1.58 ± 0.24).

In 2021, Wu and co-workers reported ^18^F-labeled radiotracers including pimonidazole derivatives [^18^F]**8,** [^18^F]**9** and [^18^F]**10**, and nitroimidazole derivatives bearing sulfonyl linkers [^18^F]**11,** [^18^F]**12** and [^18^F]**13** (Figure 7) [133]. Preliminary PET imaging studies of [^18^F]**8**, [^18^F]**9**, [^18^F]**10**, and [^18^F]**11**, using mice bearing FaDu tumors suggested that among four tracers, [^18^F]**11** exhibited the highest tumor uptakes at 0.5 h and 2 h p.i. (2.82 ± 0.66%ID/g and 2.27 ± 0.64%ID/g, respectively). In luminal-like bladder tumors (UPPL), [^18^F]**11** showed higher tumor uptake than [^18^F]**8**; moreover, at 2 h p.i., the tumor/muscle ratio of [^18^F]**11** (2.46 ± 0.48) was significantly greater compared to that of [^18^F]FMISO (1.25 ± 0.14). A radiotracer bearing 2-nitroimidazole [^18^F]**11** showed higher tumor uptake in UPPL tumors (3.36 ± 0.29%ID/g) than a radiotracer bearing 4-nitroimidazole [^18^F]**12** (1.18 ± 0.04%ID/g) and a radiotracer bearing an extending PEG linker [^18^F]**13** (3.30 ± 0.47%ID/g). In vitro stability tests showed that [^18^F]**11** was stable in plasma; however, 30 min after injecting, only 41.76% of [^18^F]**11** remained in animal blood, indicating the fast clearance of [^18^F]**11** from blood as well as the instability of [^18^F]**11** in an in vivo enzymatic environment. Comparing ex vivo autoradiography of [^18^F]**11** and immunohistochemistry of pimonidazole staining, the distribution of pimonidazole-positive regions and [^18^F]**11** were observed in similar patterns, indicating the accumulation of [^18^F]**11** in hypoxia regions in UPPL tumor tissues.

In 2022, Bernardes and co-workers developed an ^18^F-labeled analog of benzonidazole [^18^F]FBNA (N-(4-[^18^F]fluorobenzyl)-2-(2-nitro-1H-imidazol-1-yl)acetamide) ([^18^F]**15**) (Figure 8) [134]. The radiotracer ([^18^F]**15**) was prepared with 47.4 ± 5.4% RCY, >95% radiochemical purity and >40 GBq/µmol molar activity. In vitro stability tests showed that [^18^F]**15** remained intact when incubated in saline and in mouse serum. When comparing the lipophilicity of [^18^F]**15** with two hypoxia imaging probes [^18^F]FMISO and [^18^F]FAZA (log*p* = 0.36 and −0.43, respectively), [^18^F]**15** was more lipophilic (log*p* = 1.05 ± 0.04). In vitro cellular uptake studies of [^18^F]**15** were performed in AGS and the MKN45 gastric cancer cell lines. The uptake of [^18^F]**15** in AGS cells under hypoxic conditions was 4.5 times higher than under normoxic conditions. Similarly, [^18^F]**15** uptake in MKN45 cells was 4.2 times higher under hypoxic conditions than in normoxic conditions.

#### 2.1.2. ^18^F Radiotracers with Carbohydrate Structure for Hypoxia

In 2016, Kuntner and co-workers synthesized 1-(6′-deoxy-6′-[^18^F]fluoro-*β*-D- allofuranosyl)-2-nitroimidazole (*β*-6′-[^18^F]FAZAL, *β*-[^18^F]**16**) (Figure 9) [135]. *β*-[^18^F]**16** was prepared in good radiochemical purity (>98%) via the radiofluorination of a precursor which was synthesized from 1,2:5,6-di-*O*-isopropylidene-α-D-allofuranose and had a calculated ClogP value of −1.472. The binding of *β*-**16** to the nucleoside transporters SLC28A1, SLC28A3, and SLC29A1 was verified with half-maximum inhibitory concentration (IC_50_) values, which were found to be 630 ± 343, 770 ± 74, and 840 ± 22 μM, respectively. The uptake of *β*-[^18^F]**16** in the EMT6 and NCI-H1975 cell lines, cultured in vitro, showed a noticeable increase in hypoxic conditions compared to normoxic conditions, suggesting that *β*-[^18^F]**16** exhibits selectivity for hypoxic environments. At 5 h, total radioactivity uptake of *β*-[^18^F]**16** in EMT6 (4.98 ± 0.83) was higher than that in NCI-H1975 cells (3.73 ± 0.41) with increasing hypoxic/normoxic ratios during the study period. By using nucleoside transporter inhibitors, Kuntner and co-workers found that the cellular uptake of *β*-[^18^F]**16** in hypoxic tumor cells was determined by the activity of the SLC9A1 transporter. In a PET imaging study of *β*-[^18^F]**16** using mice bearing EMT6 tumor, at 120 min post injection (p.i.), tumor/muscle and tumor/blood ratios obtained using the isoflurane/air breathing protocol (2.13 ± 0.22 and 2.79 ± 0.33, respectively) were significantly higher than ratios obtained using the isoflurane/oxygen breathing protocol (1.22 ± 0.13 and 1.84 ± 0.04, respectively), indicating the hypoxic specificity of *β*-[^18^F]**16**. Ex vivo autoradiography experiments in mice bearing EMT6 tumors used to study the microtumoral distribution of *β*-[^18^F]**16** and pimonidazole, a common marker for hypoxia, showed that the high retention regions of *β*-[^18^F]**16**, positive zones in pimonidazole staining, and areas lacking blood supply in hematoxylin and eosin (HE) staining in EMT6 tumor slices were consistent, indicating the accumulation of *β*-[^18^F]**16** in tumors under hypoxic conditions.

Reischl and co-workers developed [^18^F]fluoro-azomycin-2′-deoxy-*β*-D-ribofuranoside ([^18^F]FAZDR, *β*-[^18^F]**17**) as radiotracers to mimic nucleoside structure and to improve cellular uptake through the nucleoside transporter (Figure 10) [136]. The *β*-[^18^F]**17** precursor was prepared from methyl 2-deoxy-D-ribofuranosides. *β*-[^18^F]**17** was prepared via the radiofluorination of the precursor. An in vivo PET imaging study of *β*-[^18^F]**17** and [^18^F]FAZA using BALB/c mice bearing CT26 colon carcinoma under air conditions showed significant selectivity for carcinoma uptake over muscle uptake at 1 h and 3 h. Compared to [^18^F]FAZA, *β*-[^18^F]**17** exhibited lower uptake in tumor and muscle tissues; however, at 1 h p.i., the tumor/muscle ratio of *β*-[^18^F]**17** (T/M = 1.69) was higher than that of [^18^F]FAZA (T/M = 2.76). The tumor uptake of *β*-[^18^F]**17** was inversely correlated with oxygen breathing whereas its muscle uptake exhibited no significant difference between air and oxygen breathing, indicating the selectivity of *β*-[^18^F]**17** for hypoxia. At 2 h p.i., carcinoma and muscle clearances of *β*-[^18^F]**17** showed no significant difference (54.4 ± 7.0% and 60.5 ± 8.3%, respectively) whereas the muscle clearance of [^18^F]FAZA (62.2 ± 3.1%) was significantly higher than its carcinoma clearance (32.3 ± 15.1% at 2 h p.i.), suggesting that *β*-[^18^F]**17** might be uptaken reversibly in hypoxic carcinoma and non-hypoxic muscle tissues. However, ex vivo biodistribution of *β*-[^18^F]**17** in mice bearing CT26 colon carcinoma only showed a high uptake in the intestine (indicating rapid clearance) yet a low uptake in other organs and tissues such as blood, liver, kidneys, muscle, and CT26 colon carcinomas.

In 2019, Reischl and co-workers described the synthesis of *β*-2-nitroimidazole- arabinose (*β*-FAZA) and α-2-nitroimidazole-deoxyribose (*α*-FAZDR, *α*-[^18^F]**22**) (Figure 11) [137]. Four compounds *β*-FAZA, FAZA, *α*-FAZDR and *β*-FAZDR [131] (Figure 11) were radiofluorinated with ^18^F to study the effect of configuration of 2-nitroimidazole pharmacophore and sugar moieties on the detection of hypoxia in tumors. In vitro cellular uptakes of the four compounds showed a good interaction of *β*-FAZDR with nucleoside transporters SLC28A3 and SLC28A1, and good interaction of FAZA with nucleoside transporter SLC28A1 whereas *α*-FAZDR was unable to interact with any transporter and *β*-FAZA could only inhibit transporters at high concentrations. In vivo PET imaging studies of *α*-[^18^F]**22**, *β*-[^18^F]**17** and [^18^F]FMISO in BALB/c mice bearing CT26 colon carcinoma showed that at 1 h p.i., *β*-[^18^F]**17** exhibited the highest tumor/muscle ratio (2.52 *±* 0.94) among the three radiotracers ([^18^F]FMISO: 1.37 *±* 0.11 and *α*-[^18^F]**22**: 1.93 *±* 0.39) whereas at 3 h p.i., three radiotracers gave nearly identical tumor/muscle ratios. Regarding tumor and muscle clearance, both *β*-[^18^F]**17** and *α*-[^18^F]**22** showed higher clearance rates from tumor and muscle (56.6 *±* 6.8% and 34.2 ± 7.5%, respectively) than [^18^F]FMISO (11.8 *±* 6.5%). In addition, the clearance rate of *α*-[^18^F]**22** from tumors was significantly lower than that from muscle, whereas tumor and muscle clearance rates of *β*-[^18^F]**17** were not significantly different.

Two aminooxy derivatives of 2-nitroimidazole were synthesized and radiofluorinated by Chu and co-workers in 2019 to afford radiotracers [^18^F]FDG-2NNC2ON ([^18^F]**25**) and [^18^F]FDG-2NNC5ON ([^18^F]**26**) (Figure 12) [138]. Both [^18^F]**25** and [^18^F]**26** had good stability in urine and phosphate buffer solution (PBS), and they exhibited hydrophilic properties (log*p* = −1.93 ± 0.02 for [^18^F]**25** and −1.25 ± 0.03 for [^18^F]**26**). The cellular uptake of the two radiotracers and [^18^F]FDG by S180 cells showed that there was no difference in the cellular uptakes of [^18^F]FDG under hypoxic or normoxic conditions whereas [^18^F]**25** and [^18^F]**26** exhibited 3.2 and 2.4 times higher cellular uptake under hypoxic conditions than under normoxic conditions, respectively, indicating selectivity toward hypoxia. In PET imaging studies using mice bearing S180 and OS732 tumors, [^18^F]**25** and [^18^F]**26** had low tumor uptakes ranging from 0.3 ± 0.0%ID/g to 0.4 ± 0.1%ID/g. However, owing to extremely low uptakes and fast clearance from blood and muscle, the two radiotracers still exhibited high tumor/blood ratios (3.2–3.4 for [^18^F]**25**; 4.7–7.3 for [^18^F]**26**) and tumor/muscle ratios (2.6–4.2 for [^18^F]**25**; 5.2–5.9 for [^18^F]**26**). In particular, [^18^F]**25** provided a better tumor-to-background contrast than [^18^F]**26** due to higher hydrophilicity. In addition, the obtained PET images were much clearer than those of [^18^F]FDG. The biodistribution results of the two radiotracers were consistent with the PET imaging results with high tumor/muscle and tumor/blood ratios. Moreover, co-injection of the two radiotracers with 5% glucose did not significantly change their tumor uptake values, suggesting that they did not target tumors via the glucose metabolism pathway as [^18^F]FDG. Additionally, hypoxia regions in OS732 and S180 tumors were confirmed by HIF-1α and HE staining.

### 2.2. ^99m^Tc Radiotracers for Hypoxia

^99m^Tc is a radionuclide-emitting gamma radiation widely used for SPECT imaging. ^99m^Tc possesses a favorable half-life of 6 h and low photon energy of 140 keV. In terms of convenience, compared to ^18^F, ^99m^Tc can be obtained on-site as a pertechnetate (^99m^TcO_4_^−^) by using commercial ^99^Mo/^99m^Tc generators which are smaller and more affordable than cyclotrons [139]. The preparation of ^99m^Tc-labeled complexes through coordination reactions is often conducted smoothly with high yields. However, the obtained radiolabeled products might exhibit physical and biological properties distinctly different from their precursors due to chelates. Moreover, degradation and transchelation of the ^99m^Tc-labeled complexes should be noticed because these factors might affect the stability and radiopharmaceutical applications of the complexes [140].

#### 2.2.1. ^99m^Tc Radiotracers with Mono-Nitroimidazole for Hypoxia

In 2014, Rey and co-workers synthesized ligand **L** (2-amine-3-[2-(2-methyl-5-nitro- 1H-imidazol-1-yl)ethylthio] propanoic acid) bearing a metronidazole moiety and radiolabeled it with ^99m^Tc(CO)_3_ for detecting hypoxia in tumors (Figure 13) [141]. In vitro stability studies showed that [^99m^Tc(CO)_3_(**L**)] ([^99m^Tc]**28**) remained stable in the labelling milieu and in human plasma for 4 h. Introducing a cysteine unit to the ^99m^Tc complex increased its hydrophilicity (log*p* = −0.75 ± 0.08) and decreased protein binding, which made the pharmacokinetics of the complex more appropriate for an imaging tracer. The ^99m^Tc complex with an **L** ligand was selectively uptaken by human colon adenocarcinoma cells HCT-15 in hypoxic conditions rather than in normoxic conditions (hypoxic/normoxic ratio = 1.6 ± 0.4), indicating selectivity toward hypoxia. The in vivo biodistribution in C57BL/6 mice bearing 3LL lung carcinoma cells demonstrated good tumor uptake of [^99m^Tc]**28** at 0.5 h p.i. (1.3 ± 0.4%ID/g); however, at 1 h p.i., the cellular radioactivity of [^99m^Tc]**28** was decreased by half (0.5 ± 0.1%ID/g). In general, there was a low uptake and insufficient retention of [^99m^Tc]**28** in all tissues and organs; nonetheless, the ^99m^Tc-labeled complex was rapidly eliminated from muscle, which resulted in a favorable tumor/muscle ratio of 2.0 ± 0.1%ID/g at 4 h p.i.

In 2014, Liu and co-workers developed three pegylated 2-nitroimidazole derivatives as ligands and radiolabeled them with ^99m^Tc(CO)_3_ complex to give ^99m^Tc complexes ([^99m^Tc(CO)_3_(BPA-PEG_3_-NIM)]^+^ ([^99m^Tc]**29**), [^99m^Tc(CO)_3_(AOPA-PEG_3_-NIM)] ([^99m^Tc]**30**), and [^99m^Tc(CO)_3_(IDA-PEG_3_-NIM)]^–^ ([^99m^Tc]**31**) (Figure 14) [142]. Three ^99m^Tc complexes showed good stability and had low log*P* values. In particular, [^99m^Tc]**31** was the most hydrophilic complex, followed by [^99m^Tc]**29** and neutral complex [^99m^Tc]**30** with log*P* values of −1.64 ± 0.10, −1.11 ± 0.08, and −0.44 ± 0.10, respectively. The biodistribution of three ^99m^Tc complexes in Kunming mice with S180 cancer xenografts revealed high uptake values in kidneys and livers, indicating an excretion process via renal or hepatobiliary pathways. At 2 h p.i., tumor uptakes of [^99m^Tc]**29** (0.37 ± 0.05%ID/g) and [^99m^Tc]**30** (0.40 ± 0.10%ID/g) were higher than with [^99m^Tc]**31** (0.25 ± 0.02%ID/g). Among the three ^99m^Tc complexes, [^99m^Tc]**30** exhibited the highest tumor/muscle ratio (3.01 ± 1.20 at 2 h p.i.). However, [^99m^Tc]**29**, [^99m^Tc]**30**, and [^99m^Tc]**31** showed relatively low tumor uptakes at 2 h p.i. (0.37 ± 0.05, 0.40 ± 0.10, and 0.25 ± 0.02%ID/g, respectively). The effect of chelates on physicochemical properties and tumor uptake of radiotracers was confirmed.

In 2014, Banerjee and co-workers prepared nine ^99m^Tc complexes bearing 2-, 4- or 5-nitroimidazole moieties and tridentate ligands (IDA, DETA and AEG) (Figure 15) [143]. Radiolabeling reactions of nine ligands (**32a**–**c**, **33a**–**c**, and **34a**–**c**) with [^99m^Tc(CO)_3_(H_2_O)_3_]^+^ precursor produced the corresponding ^99m^Tc(CO)_3_ in >94% RCY, >94% radiochemical purity, and >104.8 μCi/μmol specific activity. Among nine ^99m^Tc complexes, three IDA-^99m^Tc(CO)_3_ complexes (log*p* ranging from 0.39 to 0.48) were more lipophilic than DETA-^99m^Tc(CO)_3_ (log*P* ranging from 0.15 to 0.28) and AEG-^99m^Tc(CO)_3_ complexes (log*p* ranging from −0.53 to 0.06). In vivo biodistributions of nine complexes and [^18^F]FMISO in Swiss mice bearing HSDM1C1 murine fibrosarcoma showed that tumor uptake and tumor/blood ratio of [^18^F]FMISO was still higher than in the nine ^99m^Tc complexes bearing nitroimidazoles. Noticeably, the lipophilicity of the ligands significantly affected the blood activity of the complexes. In particular, nitroimidazole-IDA-^99m^Tc(CO)_3_ complexes exhibited slow clearance from blood; thus, the tumor/blood ratios were low (from 0.18 to 0.61 at 3 h p.i.). In contrast, tumor/blood ratios of nitroimidazole–DETA-^99m^Tc(CO)_3_ complexes (from 0.84 to 1.51 at 3 h p.i.) and nitroimidazole-AEG-^99m^Tc(CO)_3_ complexes (from 1.06 to 1.78 at 3 h p.i.) were improved due to rapid blood clearance. Compared to [^18^F]FMISO (T/B = 3.85 ± 0.23 at 3 h p.i.), the nine ^99m^Tc(CO)_3_-labeled complexes exhibited lower tumor/blood ratios.

In 2015, Chu and Sun reported the in vivo strain-promoted cyclooctyne-azide cycloaddition (SPAAC, click reaction) between 2-nitroimidazole-azide (2NIM-Az) as the hypoxia targeting agent and ^99m^Tc-azadibenzocyclooctyne-MAMA (^99m^Tc-AM, [^99m^Tc]**37**) for radiolabeling 2NIM-Az in vivo (Figure 16) [144]. In control experiments, [^99m^Tc]**37** was used as a blank control and ^99m^Tc-triazole-2NIM ([^99m^Tc]**38**) was used as a conventional control for hypoxia imaging. Biodistribution studies of a pretargeting method in male Kunming mice bearing S180 tumors with 2, 5, and 12 h intervals between injections of 2NIM-Az and [^99m^Tc]**37** (Figure 17) showed that pretargeting with a 5 h injection interval exhibited the highest tumor/muscle ratio (8.55 ± 0.57 at 8 h p.i.) as well as improved tumor uptake (0.70 ± 0.09%ID/g at 8 h p.i.) and tumor/blood ratio (1.44 ± 0.06 at 8 h p.i.). This indicated that a longer circulation time resulted in higher tumor uptake and tumor retention. Pretargeting methods gave similar tumor uptakes and tumor/blood ratio as [^99m^Tc]**38**. Most importantly, hypoxia pretargeting exhibited a much higher tumor/muscle ratio in comparison to [^99m^Tc]**38** (0.72 ± 0.10 at 8 h p.i.), indicating that the in vivo reaction of pretargeting azide groups and ^99m^Tc-labeled complex could detect hypoxic tumors more effectively than a conventional imaging agent. Lower tumor uptake of [^99m^Tc]**38** was due to the deactivation of 2-nitroimidazole to target hypoxia by the radiolabeled complex.

Banerjee and co-workers developed 2-nitroimidazole–dipicolylamine–^99m^Tc(CO)_3_ (2-NI–DPA–^99m^Tc(CO)_3_, [^99m^Tc]**39**) in 2016 (Figure 18) [145]. When analyzing extracted blood samples by radio-HPLC, [^99m^Tc]**39** showed good in vivo stability at 3 h p.i. In comparison to the previously reported complex 2-NI–DETA–^99m^Tc(CO)_3_ (log*P* = 0.28) [143], [^99m^Tc]**39** (log*p* = 0.38) was more lipophilic. The biodistribution of the two complexes in Swiss mice bearing fibrosarcoma tumors suggested that tumor uptake of [^99m^Tc]**39** (0.70 ± 0.16%ID/g at 3 h p.i.) was significantly higher than that of the 2-NI–DETA–^99m^Tc(CO)_3_ complex (0.24 ± 0.06%ID/g at 3 h p.i.). After 1 h p.i., the [^99m^Tc]**39** complex showed slow clearance from the tumor and rapid clearance from muscles, indicating selective retention in tumors due to the reduction of the complex and non-specific binding. As a consequence of higher lipophilicity, the [^99m^Tc]**39** complex exhibited slower clearance from normal tissues, tumors, and blood compared to the 2-NI–DETA–^99m^Tc(CO)_3_ complex. Slow blood clearance of the [^99m^Tc]**39** complex resulted in higher tumor uptake values than the 2-NI–DETA–^99m^Tc(CO)_3_ complex; however, the slow blood clearance of [^99m^Tc]**39** also led to an unfavorable tumor/blood ratio (0.83 ± 0.03 at 3 h p.i.). Although there was no apparent increase in the tumor/background ratios of [^99m^Tc]**39** compared to 2-NI–DETA–^99m^Tc(CO)_3_, [^99m^Tc]**39** still showed an improvement in tumor accumulation.

In 2016, Banerjee and co-workers prepared ^99m^Tc(CO)_3_-labeled triazole derivatives of 2-, 4-, and 5-nitroimidazole ([^99m^Tc]**40a**–**c**) for the detection of tumor hypoxia (Figure 19) [146]. The biodistribution of three ^99m^Tc(CO)_3_ complexes and [^18^F]FMISO in Swiss mice bearing fibrosarcoma tumors showed that tumor uptake values of the ^99m^Tc complexes depended on single electron reduction potential (SERP) values (indicating the ability to reduce nitroimidazole in hypoxic regions) and their lipophilicities. Among the complexes, the [^99m^Tc]**40b** exhibited the lowest tumor uptakes at the time points studied (30 min, 1 h, and 3 h), which was consistent with the lowest SERP value (−527 mV relative to standard hydrogen electrode (SHE)) of 4-nitroimidazole and the most rapid blood clearance because of the lowest lipophilicity (log*p* = −0.52 ± 0.01). Even though [^99m^Tc]**40a** had the highest SERP value (−418 mV relative to SHE), the [^99m^Tc]**40c** exhibited the highest tumor uptake values at 30 min p.i. (2.03 ± 0.32%ID/g) and at 1 h p.i. (1.45 ± 0.08%ID/g). At 3 h p.i., [^99m^Tc]**40c** exhibited a similar tumor uptake (0.81 ± 0.06 %ID/g) as [^99m^Tc]**40a** (0.75 ± 0.14%ID/g). The unexpectedly high tumor uptake of [^99m^Tc]**40c** was explained by the highest lipophilicity among the three complexes (log*p* = −0.03 ± 0.01), which led to the slowest blood clearance and longest retention in tumor cells. Due to the slow blood clearance, [^18^F]FMISO still had a higher tumor uptake value at 30 min p.i. (4.65 ± 0.86%ID/g) than [^99m^Tc]**40c** (2.03 ± 0.32%ID/g). However, the results indicated that, aside from 2-nitroimidazole, 5-nitroimidazole was also a potential moiety in radiotracers for hypoxia imaging.

In 2016, Li and co-workers reported ^99m^Tc-labeled 5-ntm-asp (3-amino-4-[2- (2-methyl-5-nitro-1H-imidazol)-ethylamino]-4-oxo-butyrate) ([^99m^Tc]**41**) (Figure 20) [147]. Two previously reported ^99m^Tc-labeled complexes containing 2-ntm-IDA and 5-ntm-IDA ligands (2- and 5-nitroimidazole-iminodiacetic acid) ([^99m^Tc]**42**, [^99m^Tc]**43**) were also synthesized. Three ^99m^Tc-labeled complexes remained stable in PBS and in human serum for a duration of 4 h. Due to the effect of the asparagine unit, [^99m^Tc]**41** had the lowest log*P* value (−0.72 ± 0.05) compared to [^99m^Tc]**42** (log*p* = 0.47 ± 0.03) and [^99m^Tc]**43** (log*p* = 0.38 ± 0.02). The three complexes also had low plasma protein binding, which allowed more unbound ^99m^Tc complexes to diffuse into tumor cells. In vitro cellular uptake studies using A549 human lung cancer cells showed that all ^99m^Tc complexes exhibited hypoxia selectivity. In particular, [^99m^Tc]**41** showed the highest cellular uptake in both hypoxic and normoxic conditions (12.58 ± 0.73 and 40.87 ± 4.74, respectively); however, its hypoxic/normoxic ratio (3.25 ± 0.08) was still lower than that of [^99m^Tc]**43** (4.47 ± 0.10). In normoxic conditions, [^99m^Tc]**41** had a higher cellular uptake than [^99m^Tc]**42** and [^99m^Tc]**43**, indicating the effect of the asparagine unit on cellular uptake status. The in vivo biodistribution in BALB/c nude mice bearing A549 cells suggested that the three ^99m^Tc-labeled complexes were excreted from the intestine and kidneys due to high uptake in these organs. Among three ^99m^Tc-labeled complexes, [^99m^Tc]**41** exhibited the highest tumor/muscle ratios at 30 min p.i. (4.65 ± 0.07) and 4 h p.i. (3.29 ± 0.07). In addition, tumor/non-tumor ratios of [^99m^Tc]**41** were always higher than that of [^99m^Tc]**43**. [^99m^Tc]**41** was considered to have the most potential among the three ^99m^Tc-labeled complexes due to appropriate lipophilicity and pharmacokinetics.

In 2017, Banerjee and co-workers reported the synthesis of two ^99m^Tc-labeled complexes, [^99m^Tc(NS_3_)(2NimNC)] ([^99m^Tc]**44**) and [^99m^Tc(NS_3_)(MetNC)] ([^99m^Tc]**45**) bearing ‘4 + 1’ mixed-ligands (Figure 21) [148]. The SERP of [Re(NS_3_)(2NimNC)], [Re(NS_3_)(MetNC)] and the corresponding nitroimidazole ligands using cyclic voltammetry indicated that the metal complex did not affect the ability of nitroimidazole ligands to be reduced by hypoxic cells. Both [^99m^Tc]**44** and [^99m^Tc]**45** were highly lipophilic with log*p* values of 0.94 ± 0.1 and 0.97 ± 0.07, respectively. Biodistribution in Swiss mice bearing fibrosarcoma tumors showed fast clearance of the ^99m^Tc-labeled complexes from blood and high uptakes in the liver and intestine due to high lipophilicity. The low retention of the ^99m^Tc-labeled complexes in blood resulted in low tumor uptakes (0.34 ± 0.03%ID/g at 3 h p.i. for [^99m^Tc]**44** and 0.31 ± 0.05%ID/g at 3 h p.i. for [^99m^Tc]**45**). The tumor uptake values of both complexes were significantly lower than the less lipophilic tracer [^18^F]FMISO (log*p* = −0.41; tumor uptake = 2.04 ± 0.14%ID/g). Tumor/muscle ratios of [^99m^Tc]**44** and [^99m^Tc]**45** reached their maximum values (4.76 ± 1.08 and 4.84 ± 1.49, respectively) at 3 h p.i., and showed no significant difference compared to [^18^F]FMISO (4.84 ± 0.90 at 3 h p.i.). Tumor/blood ratios of [^99m^Tc]**44** and [^99m^Tc]**45** also reached maximum values (1.24 ± 0.08 and 0.89 ± 0.07, respectively) at 3 h p.i.; however, [^18^F]FMISO still exhibited a higher tumor/blood ratio (3.85 ± 0.23 at 3 h p.i.).

In 2020, ^99m^Tc-2-MBI ([^99m^Tc]**46**) was reported by Zhang and co-workers (Figure 22) [149]. [^99m^Tc]**72** complex had good stability in vitro in saline and in mice serum for 24 h. The complex exhibited lipophilicity with a log*p* value of 1.512. According to an electrophoresis study, [^99m^Tc]**46** was neutral, suggesting high compatibility with living tissues. Comparing cellular uptake values using S180 cells of [^99m^Tc]**46** in hypoxic and aerobic conditions, the ^99m^Tc-labeled complex exhibited selectivity for hypoxia with significantly higher accumulation in hypoxic cells than in aerobic cells during the period studied (6 h). Biodistribution studies in Balb/c mice bearing S180 tumor showed that [^99m^Tc]**46** exhibited higher accumulation in tumors compared to normal tissues at different time points from 30 min to 24 h. The ^99m^Tc-labeled complex exhibited the highest tumor uptake at 4 h p.i. (12.94 ± 2.09%ID/g); however, at 24 h p.i. tumor/muscle ratio (4.14 ± 0.77) and tumor/blood ratio (3.91 ± 0.63) were found to be the highest. Metabolic stability study of [^99m^Tc]**72** in mice showed intact ^99m^Tc-labeled complex in blood and urine; however, the ^99m^Tc-complex decomposed in the liver, indicating metabolism via the hepatobiliary system and excretion via the urinary system, which was also consistent with the biodistribution results of [^99m^Tc]**46**. Scintigraphy imaging studies in mice bearing S180 tumors also showed a high accumulation of [^99m^Tc]**46** in tumors with a tumor/normal tissue ratio of 3.39 ± 0.38, which were in accordance with the biodistribution studies of [^99m^Tc]**46**.

In 2021, three ^99m^Tc-labeled complexes (^99m^Tc-tricine-TPPTS-HYNICNM ([^99m^Tc]**47**), ^99m^Tc-tricine-TPPMS-HYNICNM ([^99m^Tc]**48**), and ^99m^Tc-(tricine)_2_-HYNICNM ([^99m^Tc]**75**)) bearing 6-hydrazinonicotinamide (HYNIC) 2-nitroimidazole derivative (HYNICNM) ligand (**50**) were developed by Zhang and co-workers (Figure 23) [150]. In vitro studies showed that the three ^99m^Tc-labeled complexes remained stable in saline and in mouse serum, and all were hydrophilic. In particular, [^99m^Tc]**47** was more hydrophilic (log*p* = −3.02 ± 0.08) than [^99m^Tc]**48** (log*p* = −0.76 ± 0.03) and [^99m^Tc]**49** (log*p* = −1.73 ± 0.02), indicating the effect of the co-ligand on the hydrophilicity of the complex. In vitro cellular uptake studies of the three complexes using S180 cells suggested significantly higher cellular uptake values in hypoxic conditions than in aerobic conditions at all the studied time points (1, 2, and 4 h), indicating selectivity toward hypoxia. The biodistribution of the three ^99m^Tc-labeled complexes in Kunming mice bearing S180 tumors showed high uptake values in kidneys, suggesting excretion mainly via the urinary pathway. Among the three ^99m^Tc-labeled complexes, [^99m^Tc]**47** exhibited the highest tumor uptake value (1.05 ± 0.27%ID/g at 2 h p.i.). Therefore, this complex was chosen for SPECT/CT imaging studies. In SPECT/CT imaging studies, at 2 h p.i., the uptake of [^99m^Tc]**47** was clearly observed in S180 tumors. High uptakes of [^99m^Tc]**47** in other tissues were consistent with the biodistribution results of this ^99m^Tc-labeled complex.

In 2022, Su and Chu reported four cyclopentadienyl ^99m^Tc(CO)_3_ complexes containing 2-nitroimidazole moieties in which 2-nitroimidazoles and cyclopentadienyls were linked via carbon chains of different lengths (Figure 24) [151]. Among the four ^99m^Tc-labeled complexes, [^99m^Tc]**51d** containing two 2-nitroimidazole groups was the most lipophilic (log*p* = 0.66 ± 0.02); thus, it could enter the cell more easily than the three complexes containing one 2-nitroimidazole group, and exhibited higher in vitro cellular uptake in hypoxic cells (40.0 ± 2.0%). Extending the carbon chains in the three complexes [^99m^Tc]**51a**–**c** resulted in higher lipophilicity and easier diffusion through the cell membrane, yet low cellular uptake values and low hypoxic selectivity. [^99m^Tc]**51d** showed the best selectivity toward hypoxia among the four complexes (hypoxic/aerobic uptake ratio = 1.33 at 3 h), followed by [^99m^Tc]**51a** (1.30 at 3 h), [^99m^Tc]**51b** (1.11 at 3 h), and [^99m^Tc]**51c** (0.87 at 3 h). Biodistribution studies in mice bearing S180 tumors suggested that among the three complexes containing one 2-nitroimidazole group, at 4 h p.i., the order of tumor/muscle ratios was [^99m^Tc]**51a** > [^99m^Tc]**51b** > [^99m^Tc]**51c**, whereas the order of tumor/blood ratios was [^99m^Tc]**51c** > [^99m^Tc]**51b** > [^99m^Tc]**51a**. However, at 2 h p.i., [^99m^Tc]**51b** exhibited the highest tumor uptake value among the three complexes of 2.23 ± 0.24%ID/g. [^99m^Tc] **51d** exhibited significantly higher tumor uptake (3.19 ± 0.77%ID/g at 2 h p.i.) and tumor/muscle ratio (3.47 ± 0.47 at 2 h p.i.) than [^99m^Tc]**51b**. However, the tumor/blood ratio of [^99m^Tc]**51d** was low due to high blood uptake. SPECT imaging studies of [^99m^Tc]**51b** and [^99m^Tc]**51d** showed observable radioactivity in tumors. Accumulations of [^99m^Tc]**51b** and [^99m^Tc]**51d** in tumors and organs were in accordance with biodistribution studies.

#### 2.2.2. ^99m^Tc Radiotracers with Di-Nitroimidazole for Hypoxia

In 2015, Zhang and co-workers prepared ligand 3-(4-nitro-1H-imidazolyl)propyl dithiocarbamate (N4IPDTC) and radiolabeled it with ^99m^TcN, ^99m^TcO, and ^99m^Tc(CO)_3_ (Figure 25) [152]. The radiotracers ^99m^TcN-N4IPDTC ([^99m^Tc]**53**), ^99m^TcO-N4IPDTC ([^99m^Tc]**54**), and ^99m^Tc(CO)_3_-N4IPDTC ([^99m^Tc]**55**) exhibited good in vitro stability in mouse serum for 6 h at 37 °C. Among the three ^99m^Tc complexes, [^99m^Tc]**55** was the most hydrophilic (log*p* = −0.71 ± 0.01) whereas [^99m^Tc]**53** was the most lipophilic (log*p* = 0.60 ± 0.01). The in vitro cellular uptake of the ^99m^Tc-labeled complexes using S180 cells under hypoxic and aerobic conditions showed that they were selectively uptaken in hypoxic cells. Biodistribution studies in mice bearing S180 tumors showed low muscle uptakes and good tumor uptakes of the three complexes which led to high tumor/muscle ratios. [^99m^Tc]**54** exhibited the highest tumor uptake (2.84 ± 0.19%ID/g at 2 h p.i.), tumor/muscle and tumor/blood ratios (4.44 and 1.49 at 2 h p.i., respectively) among the three ^99m^Tc complexes. The ^99m^Tc complexes were excreted mostly via renal and hepatobiliary routes. In comparison to ^99m^Tc-N4IPA (tumor uptake = 0.34 ± 0.06%ID/g), a potential hypoxia imaging agent [153], [^99m^Tc]**54** exhibited an 8-fold higher tumor uptake (2.63 ± 0.35%ID/g at 4 h p.i.). However, ^99m^Tc-N4IPA had a two times higher tumor/muscle ratio (8.60 at 4 h p.i.) than [^99m^Tc]**54** (3.87 at 4 h p.i.) because [^99m^Tc]**54** had a high uptake in the liver and kidneys. These biodistribution results were consistent with SPECT imaging of [^99m^Tc]**54** in animals.

In 2015, ^99m^Tc-Ethylenedicysteine-bis-misonidazole (^99m^Tc-EC-MISO, [^99m^Tc]**58**) was developed by Chen and co-workers (Figure 26) [154]. ^99m^Tc-EC and [^99m^Tc]**58** remained stable for 4 h in mouse serum. [^99m^Tc]**58** (log*p* = 0.85 ± 0.03) was more lipophilic than ^99m^Tc-EC (log*p* = −0.23 ± 0.21) and [^18^F]FMISO (log*p* = 0.4). Therefore, [^99m^Tc]**58** required a longer length of time after injection due to the long retention of [^99m^Tc]**58** in normal tissues, especially in the intestines. The in vivo tissue distribution of [^99m^Tc]**58** in mice bearing C6 glioma tumor showed the highest uptake value in kidneys, indicating renal excretion. Accumulation of [^99m^Tc]**58** in tumors was observed with steady tumor uptake values during the study period (1.24–1.29 from 0.5 to 4 h p.i.). The tumor/muscle ratio also increased during the study period and reached 4.68 at 4 h p.i. Autoradiography of C6 glioma tumor slices showed that [^99m^Tc]**58** exhibited a 7.7-fold stronger signal intensity than ^99m^Tc-EC at 2 h p.i. and an 8.6-fold one at 4 h p.i. Similarly, SPECT/CT images displayed a noticeable accumulation of [^99m^Tc]**58** around the tumor whereas no accumulation of ^99m^Tc-EC was detected.

A ^99m^TcN(PNP)-complex bearing metronidazole isocyanide (MetroNC) ligand was developed by Banerjee and co-workers in 2016 (Figure 27) [155]. MetroNC-[^99m^TcN(PNP)] ([^99m^Tc]**59**) was synthesized from [MetroNC] and precursor [^99m^TcN(PNP)]^2+^ with two proposed structures. [^99m^Tc]**59** complex was stable at room temperature and remained unchanged in vitro in human serum for 1 h. The in vivo biodistribution in Swiss mice bearing fibrosarcoma tumors showed that [^99m^Tc]**59** exhibited a tumor uptake value of 0.5 ± 0.04%IA/g at 3 h p.i. Compared to the ^99m^TcN-metronidazole radiotracers reported previously, [^99m^Tc]**59** exhibited lower tumor uptake than ^99m^Tc-MNZ-xanthate (1.36 ± 0.29%IA/g at 3 h p.i.) [156] and ^99m^TcN-metronidazoleDTC1 (1.00 ± 0.20%IA/g at 4 h p.i.) [157]. However, the tumor/blood ratio of [^99m^Tc]**59** (1.36 ± 0.14 at 3 h p.i.) was higher than the previously reported ^99m^TcN-metronidazole radiotracers (0.62 at 3 h p.i. for ^99m^Tc-MNZ-xanthate; 1.15 ± 0.32 at 4 h p.i. for ^99m^TcN-metronidazoleDTC1; 0.28 ± 0.07 at 4 h p.i. for ^99m^TcN-metronidazoleDTC2). The improved tumor/blood ratio value was explained by the rapid clearance of [^99m^Tc]**59** from blood and the high uptake in the liver and intestine due to high lipophilicity (log*p* = 1.1) [158]. The tumor/muscle ratio of [^99m^Tc]**59** complex showed no apparent change during the study and reached 1.68 ± 0.13 at 3 h p.i.

The EDTA derivative of 4-nitroimidazole (EDTA-4-EtNHNM, **60**) was synthesized and radiolabeled with ^99m^Tc to provide ^99m^Tc-EDTA-4-EtNHNM ([^99m^Tc]**60**) by Zhang and co-workers in 2017 (Figure 28) [159]. ^99m^Tc-EDTA was also prepared using the method reported by Cash and co-workers in 1980 [160]. In in vitro stability tests, [^99m^Tc]**60** exhibited stability in both the labeling milieu and mouse serum, and had low lipophilicity (log*p* = −1.07 ± 0.05). In vitro cellular uptake studies using S180 cell lines showed that cellular uptakes of [^99m^Tc]**60** in anoxic conditions were significantly higher than those in normoxic conditions, indicating hypoxia selectivity. The control complex ^99m^Tc-EDTA, however, exhibited no preference for hypoxia. In biodistribution studies in S180 tumor-bearing Kunming mice, [^99m^Tc]**60** exhibited a higher tumor uptake (1.38 ± 0.22%ID/g) and lower kidney and stomach uptakes compared to ^99m^Tc-EDTA due to its hydrophilicity. Also, at 2 h p.i., [^99m^Tc]**60** exhibited a relatively high tumor/muscle ratio of 3.53 and tumor/blood ratio of 1.04 because of the slow clearance from tumor yet fast clearance from muscle and blood. SPECT imaging results were also consistent with biodistribution results, both ^99m^Tc complexes were observed in tumor regions. However, [^99m^Tc]**60** still had a higher region of interest (ROI) ratio (3.30) than ^99m^Tc-EDTA (ROI ratio = 1.62).

To enhance the tumor/blood ratio of the previously reported complex ^99m^Tc-EDTA-4-EtNHNM [159], Zhang and co-workers synthesized the ^99m^Tc-complex ^99m^Tc-EDTA-2-EtNHNM ([^99m^Tc]**61**) by using a similar procedure (Figure 29) [161]. [^99m^Tc]**61** showed good stability in vitro in both the labeling milieu and mouse serum. Compared to ^99m^Tc-EDTA bearing 4-nitroimidazole (log*p* = −1.07 ± 0.05), [^99m^Tc]**61** was more hydrophilic (log*p* = −1.93 ± 0.02). During the experimental period of the in vitro cellular uptake study, [^99m^Tc]**61** exhibited a higher uptake in hypoxic than in normoxic conditions, indicating selectivity toward hypoxia. Biodistribution studies of [^99m^Tc]**61** using mice bearing S180 tumors did not show a significant improvement in tumor uptake (1.17 ± 0.23%ID/g at 4 h p.i.) when compared to ^99m^Tc-EDTA-4-EtNHNM (1.04 ± 0.15%ID/g at 4 h p.i.). However, the fast clearance of [^99m^Tc]**61** from muscle resulted in a higher tumor/muscle ratio (6.27 at 4 h p.i.). Importantly, the tumor/blood ratio of [^99m^Tc]**61** was significantly improved (2.02 at 4 h p.i.) because the clearance from blood was faster than the clearance from tumors, which helped the ^99m^Tc-labeled complex prolong retention time in tumors. SPECT images of [^99m^Tc]**61** also showed consistent results with the biodistribution studies. The obvious accumulation at tumor sites was indicated by an ROI tumor/non-tumor ratio of 5.92.

In 2018, Zhang and co-workers synthesized the xanthate derivative of secnidazole (SNXT) as a didentate ligand and radiolabeled the ligand with [^99m^TcN]^2+^ and [^99m^TcO]^3+^ precursors (Figure 30) [162]. The ^99m^TcN-SNXT and ^99m^TcO-SNXT complexes ([^99m^Tc]**62**, [^99m^Tc]**63**) were both hydrophilic with log*P* values below zero (−0.83 ± 0.17 for [^99m^Tc]**62** and −0.84 ± 0.04 for [^99m^Tc]**63**) and showed good stability in the reaction mixture and in mouse serum. The in vitro cellular uptakes of the two complexes using an S180 tumor cell line were higher in hypoxic conditions than in normoxic conditions, indicating selectivity for hypoxia. Biodistribution studies in mice bearing S180 tumors showed similar initial tumor uptake values due to the similar hydrophilicity of the two ^99m^Tc-labeled complexes. [^99m^Tc]**63** retained its uptake values in tumor tissue at 0.5 h p.i. (1.89 ± 0.13%ID/g) and at 4 h p.i. (1.85 ± 0.06%ID/g). On the other hand, the tumor uptake of [^99m^Tc]**62** at 0.5 h p.i. was significantly reduced from 1.97 ± 0.64%ID/g to 1.20 ± 0.30%ID/g at 4 h p.i. Both ^99m^Tc-labeled complexes showed high tumor/muscle ratios due to the fast clearance from tumors; in particular, [^99m^Tc]**63** exhibited a higher tumor/muscle ratio (4.18 at 4 h p.i.) compared to that of [^99m^Tc]**62** (1.36 at 4 h p.i.). In addition, SPECT images of [^99m^Tc]**63** were consistent with the biodistribution results with observable tumor uptake and high accumulation in the liver and kidneys.

In 2020, Zhang and co-workers developed two ^99m^Tc-labeled complexes (^99m^TcN-NMXT ([^99m^Tc]**65**) and ^99m^TcO-NMXT([^99m^Tc]**66**)) containing a 4-nitroimidazole xanthate ligand (NMXT, **71**) (Figure 31) [163]. Two ^99m^Tc complexes were both stable in saline and in mouse serum. Both [^99m^Tc]**65** and [^99m^Tc]**66** were hydrophilic and had similar log*P* values (−0.66 ± 0.02 and −0.75 ± 0.06, respectively). In vitro cellular uptake studies using S180 cell lines showed that the two ^99m^Tc-labeled complexes exhibited good selectivity for hypoxia and cellular uptakes in hypoxic conditions were significantly higher than those in aerobic conditions. Biodistribution of [^99m^Tc]**65** and [^99m^Tc]**66** in Kunming mice bearing S180 tumors showed that [^99m^Tc]**66** exhibited higher tumor uptake (1.93 ± 0.25%ID/g at 2 h p.i.) than [^99m^Tc]**65** (0.90 ± 0.27 at 2 h p.i.) due to high blood uptake and slow blood clearance. However, the higher blood uptake of [^99m^Tc]**66** resulted in a lower tumor/blood ratio (0.83 ± 0.05 at 2 h p.i.) compared to [^99m^Tc]**65** (1.32 ± 0.27 at 2 h p.i.). On the other hand, [^99m^Tc]**66** showed a higher tumor/muscle ratio (5.33 ± 0.24 at 2 h p.i.) than [^99m^Tc]**65** (1.06 ± 0.35 at 2 h p.i.). SPECT/CT imaging results of [^99m^Tc]**66** were consistent with the biodistribution results; in particular, at 2 h p.i., accumulation of [^99m^Tc]**66** was observed in the tumor with an ROI ratio of 6.15 ± 0.97.

In 2023, Li and Chu synthesized three ^99m^Tc-labeled complexes (^99m^Tc-2P2O1 ([^99m^Tc]**68a**), ^99m^Tc-2P2O2 ([^99m^Tc]**68b**), and ^99m^Tc-2P2O4 ([^99m^Tc]**68c**)) [164] by using PEG modifications of the previously reported nitroimidazole propylene amine oxime (PnAO) complexes ^99m^Tc-2P2 [165] to improve its pharmacokinetic properties (Figure 32). Partition coefficients of PEG-modified complexes (from 0.98 ± 0.01 to 1.14 ± 0.21) were not significantly different from that of ^99m^Tc-2P2 (1.09 ± 0.12), indicating that introducing ethylene glycol units had no effect on the lipophilicity. The three complexes remained intact in vitro in PBS and fetal bovine serum (FBS), and in vivo in normal mice. In vitro cellular uptake studies using an S180 cell line suggested that the three complexes were selectively uptaken under hypoxic conditions with hypoxic/normoxic ratios from 2.29 ± 0.67 to 2.92 ± 0.61. Biodistribution studies in mice bearing S180 tumors showed no significant difference in tumor uptake values at 4 h p.i. of the three ^99m^Tc-labeled complexes (from 0.71 ± 0.14 to 1.00 ± 0.26%ID/g) and ^99m^Tc-2P2 (0.86 ± 0.22%ID/g). However, the tumor/muscle ratios of PEG-modified complexes (from 5.56 ± 1.10 to 7.20 ± 2.37) were significantly higher than those of ^99m^Tc-2P2 (T/M = 3.24 ± 0.65 at 4 h p.i.). Additionally, the three PEG-modified complexes also exhibited improved tumor/blood ratios (from 1.66 ± 0.34 to 2.13 ± 0.19) compared to that of ^99m^Tc-2P2 (0.81 ± 0.34) [166], indicating the effect of adding an ethylene glycol chain into the structures. In particular, among the three PEG-modified complexes, [^99m^Tc]**68c** demonstrated the highest tumor/blood ratio at 4 h p.i. (2.13 ± 0.19). Autoradiography imaging studies showed that [^99m^Tc]**68c** had a heterogenous distribution in S180 tumors. These results are consistent with positive regions in immunohistochemical staining of H1F-1α which confirmed the selective tumor hypoxia targeting effect of [^99m^Tc]**68c**.

#### 2.2.3. ^99m^Tc Radiotracers with Multi-Nitroimidazole for Hypoxia

In 2018, Mallia and co-workers synthesized a ^99m^Tc(CO)_3_ complex bearing three metronidazole isocyanide moieties (MetroNC) ([^99m^Tc]**69**) (Figure 33) [167]. The MetroNC ligand was prepared by using a similar method to the preparation of MetroNC-[^99m^TcN(PNP)] [155]. The cyclic voltammetry experiment indicated that the metal center in the Re(CO)_3_(MetroNC)_3_ complex had a slight effect on the reduction potential (SERP value), with a value of −0.90 V, compared to the MetroNC ligand alone, which had a value of −0.96 V. In vitro studies of [^99m^Tc]**69** showed that the ^99m^Tc-labeled complex had good stability in human serum for 3 h, and was lipophilic (log*p* = 0.46 ± 0.04). When incubated in CHO cells in hypoxic or normoxic conditions, [^99m^Tc]**69** complex showed an increasing and selective accumulation in hypoxic conditions with a hypoxic/normoxic ratio > 2. Biodistribution studies in Swiss mice bearing fibrosarcoma showed a good initial uptake and retention of [^99m^Tc]**69** complex in tumors at 30 min and 1 h p.i. (0.67 ± 0.10 and 0.67 ± 0.17%ID/g, respectively). At 3 h p.i., tumor uptake was only maintained (0.31 ± 0.09%ID/g). Blood and muscle uptakes of [^99m^Tc]**69** decreased during the period studied. Calculated tumor/blood and tumor/muscle ratios were maximum at 1 h p.i. (1.00 ± 0.27 and 4.38 ± 0.51, respectively).

In 2020, Zhang and co-workers developed two ^99m^Tc(CO)_3_ complexes bearing isocyanide derivative of 4-nitroimidazole (**73**) (Figure 34) [168] by using a synthetic pathway reported by Denny and co-workers in 1994 [169]. Both [^99m^Tc]**71** and [^99m^Tc]**72** had good in vitro stability in saline and in mouse serum but [^99m^Tc]**71** was more hydrophilic (log*p* = −1.22 ± 0.03) than [^99m^Tc]**72** (log*p* = −0.72 ± 0.03). Both ^99m^Tc-labeled complexes exhibited higher uptake values in S180 tumor cells in hypoxic conditions than in aerobic conditions, indicating selectivity for hypoxia. Both ^99m^Tc(CO)_3_ complexes showed fast clearance from blood due to hydrophilicity. In particular, [^99m^Tc]**71** was cleared from blood more quickly than [^99m^Tc]**72** because of a lower log*P* value. Noticeably, fast blood clearance could cause low tumor uptake due to a lack of time for passive diffusion. Hence, [^99m^Tc]**71** had a lower tumor uptake (0.22 ± 0.03%ID/g at 2 h p.i.) than [^99m^Tc]**72** (0.71 ± 0.10%ID/g at 2 h p.i.). However, because the muscle uptake of [^99m^Tc]**71** was lower than its tumor uptake, tumor/muscle ratios were still relatively high (3.65 at 2 h p.i.) and higher than those of [^99m^Tc]**72** (2.40 at 2 h p.i.). Tumor/blood ratios of [^99m^Tc]**71** (1.00 at 2 h p.i.) were also higher than those of [^99m^Tc]**72** (0.56 at 2 h p.i.).

In 2018, Zhang and co-workers developed four ^99m^Tc-labeled complexes containing 2-nitroimidazole isocyanide derivatives (Figure 35) [170]. Four complexes [^99m^Tc]**74a**–**d** exhibited stability in both the labeling milieu and mouse serum and exhibited hydrophilicity. In particular, log*P* values increased when the number of CH_2_ groups increased from two groups in [^99m^Tc]**74a** (−2.65 ± 0.14) to five groups in [^99m^Tc]**74d** (−0.45 ± 0.05). In vitro cellular uptake studies of four ^99m^Tc-labeled complexes using S180 cells showed significantly higher cellular uptakes in hypoxic than aerobic conditions, suggesting selectivity for hypoxia. The biodistribution of four [^99m^Tc]**74a**–**d** complexes in Kunming mice bearing S180 tumors showed relatively high tumor uptakes and low muscle uptakes at 2 h p.i., resulting in high tumor/muscle ratios. Among the four ^99m^Tc-labeled complexes, [^99m^Tc]**74c** exhibited the highest tumor uptake (0.83 ± 0.14%ID/g at 2 h p.i.) and the highest tumor/muscle ratio (5.05 at 2 h p.i.). SPECT/CT images of [^99m^Tc]**74c** were in accordance with its biodistribution results with an observable accumulation of [^99m^Tc]**74c** in tumor regions (ROI ratio = 5.39 ± 0.67). The high accumulation of this complex was also observed in the liver and kidneys.

In 2020, Zhang and co-workers synthesized 4-nitroimidazole isocyanide derivative **M** (**78**) and radiolabeled it with [^99m^Tc(I)]^+^ and [^99m^Tc(I)(CO)_3_]^+^ cores (Figure 36) [171]. Both [^99m^Tc(**M**)_6_]^+^ ([^99m^Tc]**76**) and [^99m^Tc(CO)_3_(**M**)_3_]^+^ ([^99m^Tc]**77**) showed good stability in saline and in mouse serum. Both complexes were hydrophilic but [^99m^Tc]**76** (log*p* = −2.21 ± 0.06) had a higher hydrophilicity than [^99m^Tc]**77** (log*p* = −1.22 ± 0.03). In vitro cellular uptake studies using S180 cells showed that both complexes exhibited higher uptake in hypoxic conditions than in aerobic conditions. In addition, [^99m^Tc]**76** showed higher in vitro cellular uptake than [^99m^Tc]**77** due to more 4-nitroimidazole pharmacophores. Biodistribution results of [^99m^Tc]**76** and [^99m^Tc]**77** in Kunming mice bearing S180 tumors were consistent with the results of in vitro cellular uptake studies. [^99m^Tc]**76** exhibited higher tumor uptake values than [^99m^Tc]**77** due to bearing more **M** ligands. The highly hydrophilic [^99m^Tc]**76** complex showed a fast clearance from normal tissues; therefore, [^99m^Tc]**76** exhibited higher tumor/blood, tumor/muscle, and tumor/liver ratios than [^99m^Tc]**77** at 2 h p.i. Due to its higher hydrophilicity, [^99m^Tc]**76** was mainly excreted by the kidneys whereas [^99m^Tc]**77** was mainly excreted by the liver. SPECT/CT imaging studies showed that the accumulation of the two complexes in tumors was observable and [^99m^Tc]**76** exhibited a higher ROI ratio (5.64 ± 0.64) than [^99m^Tc]**77** (3.49 ± 0.33) at 2 h p.i., indicating accordance with the biodistribution results.

### 2.3. ^131^I-/^125^I Radiotracers for Hypoxia

Similar to ^18^F, radioactive iodine can also be directly incorporated into various molecules while retaining its biological properties. Radionuclide ^124^I has a long half-life of 4 days which enables distant distribution as well as long-term PET imaging studies [172,173]. ^131^I emits beta irradiation and is used for the preparation of SPECT imaging agents. ^125^I, a SPECT radionuclide-emitting photon irradiation, has a much longer half-life than ^124^I (59.5 days) [174]. This long half-life of ^125^I might not be favorable for in vivo SPECT imaging due to the long exposure of the body to radiation. As a radioactive halogen like ^18^F, the yields of radioiodination reactions might not always be high, limiting the application of radioactive iodine. Additionally, iodine radionuclides are produced using a cyclotron, which is another drawback.

In 2019, Chu and co-workers developed ^131^I- and ^125^I-labeled radiotracers containing 2-nitroimidazole and 2-(4′-aminophenyl)benzothiazole (BTA) ([^131^I]I2NP1BTA ([^131^I]**80**) and [^131^I]I2NP2BTA ([^131^I]**81**) (Figure 37) [175]. To improve the pharmacokinetics of the previously synthesized radiotracer [^131^I]2NPBTA ([^131^I]**79**) [176], a PEG chain was introduced into the radiotracers as a linker to connect 2-nitroimidazole and the BTA group. In vitro studies showed that [^131^I]**80** and [^131^I]**81** remained stable in PBS and FBS for 8 h and were both lipophilic. In particular, [^131^I]**80** (log*p* = 1.57 ± 0.07) was more lipophilic than [^131^I]**81** (log*p* = 1.17 ± 0.09) due to a shorter PEG chain. In vitro cellular uptake studies of two radiotracers indicated higher cellular uptakes in hypoxic conditions than in normoxic conditions at 2 h p.i.; however, the obtained cellular uptake values in hypoxic cells were not high. Biodistribution studies of [^131^I]**80** and [^131^I]**81** in mice bearing S180 tumors showed the highest tumor uptake values at 1 h p.i (1.19 ± 0.13%ID/g and 1.05 ± 0.21%ID/g, respectively). There was no significant change in tumor/muscle ratios of the two radiotracers during the period studied (2.33 ± 0.65%ID/g for [^131^I]**80** and 2.08 ± 0.58%ID/g for [^131^I]**81**, at 6 h p.i.). In addition, at 6 h p.i., [^131^I]**80** had a higher tumor/blood ratio than [^131^I]**81**; therefore, [^131^I]**80** might provide better resolution in imaging than [^131^I]**81**. The uptake of the two radiotracers was high in the kidney, suggesting excretion via the urinary system. SPECT/CT image study showed the obvious accumulation of [^125^I]**80** and [^125^I]**81** in tumors and in kidneys, in accordance with the biodistribution of the two radiotracers.

In 2020, to study how introducing a second 2-nitroimidazole group in the structure of radiotracers affected the detection of hypoxia, Chu and co-workers synthesized eight ^131^I-radiolabeled 2-nitroimidazole derivatives in which [^131^I]**82** and [^131^I]**83** contained one 2-nitroimidazole moiety and [^131^I]**84**–**89** contained two 2-nitroimidazole moieties (Figure 38) [177]. Of the eight ^131^I-labeled radiotracers, [^131^I]**82** was the most lipophilic (log*p* = 1.54 ± 0.03) and [^131^I]**83** was the most hydrophilic (log*p* = −0.70 ± 0.02). Introducing a benzene group to the compound increased the log*P* values whereas introducing a second 2-nitroimidazole group to the molecule decreased the log*p* values of the radiotracers. In vitro cellular uptakes of eight ^131^I-labeled radiotracers using S180 tumor cells showed that [^131^I]**83**–**86** had high hypoxic/normoxic ratios. [^131^I]**85** exhibited a significantly higher hypoxic/normoxic ratio (4.4 ± 0.5) than other ^131^I-labeled radiotracers because its proper log*p* value allowed for diffusion across the cell membrane and the two 2-nitroimidazole groups could target hypoxia. The biodistribution of [^131^I]**83**–**86** in Kunming mice bearing S180 tumors suggest that [^131^I]**85** exhibited the most suitable tumor/blood ratio of 2.03 ± 0.45 and tumor/muscle ratio of 6.82 ± 1.70 at 8 h p.i. In comparison to [^19^F]FMISO (T/M = 1.4 ± 0.4 at 2 h p.i., T/B = 2.8 ± 0.4 at 2 h p.i.), [^131^I]**85** exhibited a much higher tumor/muscle ratio (6.82 ± 1.70 at 4 h p.i.) but a slightly lower tumor/blood ratio (2.03 ± 0.45 at 4 h p.i.).

### 2.4. ^64^Cu Radiotracers for Hypoxia

^64^Cu is one of the copper radioisotopes used in molecular imaging and radiotherapy. With a half-life of 12.7 h, many ^64^Cu-labeled radiotracers were developed for the PET imaging of many types of cancers [178]. The production of ^64^Cu-labeled radiotracers is mainly carried out via the coordination of a variety of chelators to ^64^Cu, which can be obtained from a cyclotron or reactor. Similar to ^99m^Tc, the coordination of ^64^Cu might be unstable, which will affect the stability of the radiotracer in vivo. In addition, the inconvenience of using a cyclotron to generate ^64^Cu is also another drawback of ^64^Cu-labeled radiotracers.

In 2016, Yang and co-workers reported a ^64^Cu-labeled complex containing bis(2-nitroimidazole) (^64^Cu-BMS2P2, [^64^Cu]**90**) (Figure 39) [179]. In vitro stability studies showed that [^64^Cu]**90** remained stable (>90% radiochemical purity) in saline for 60 h. An in vitro cellular uptake study of [^64^Cu]**90** using HUH-7 cancer cells showed that, at different time points from 1 h to 4 h, and [^64^Cu]**90** exhibited significantly higher cellular uptakes in hypoxic conditions than in normoxic conditions. The hypoxic/normoxic ratio of [^64^Cu]**90** was highest at 3 h (2.59). In addition, the cellular uptake of [^64^Cu]**90** in hypoxic conditions was higher than that of ^64^Cu-BMS181321 containing one nitroimidazole group [180], indicating that adding a second nitroimidazole group increased cellular uptake in hypoxic conditions. An in vivo PET imaging study of [^64^Cu]**90** using mice bearing A549 tumors showed a clearly observed uptake of [^64^Cu]**90** in tumors, intestines, and liver. CA9 immunohistochemistry demonstrated a high expression of CA9 in tumors and the expression of CA9 was also consistent with the results of PET imaging, indicating the hypoxic specificity of [^64^Cu]**90**.

## 3. Conclusions and Perspectives

Hypoxia, a low level of oxygen, is a common feature in solid tumors. Tumor hypoxia has been considered a negative factor in the treatment of cancer due to the resistance to radiotherapy and chemotherapy it causes. Therefore, an accurate assessment of the hypoxia status of tumors before aggressive cancer treatments is crucial to reduce poor outcomes and mortality.

Molecular imaging methods for the detection of tumor hypoxia have received growing attention due to their non-invasive nature, repeatability, uniformity, and ability to detect biological processes in vivo [181]. In recent years, many radiotracers for targeting hypoxia were studied in vitro on hypoxic cancer cell lines and in vivo in animals bearing hypoxic tumors.

[^18^F]FMISO, a hypoxia marker for PET imaging, has been used to evaluate hypoxia in many clinical studies [182] and was also involved as a control for hypoxia in many studies. However, it still has several drawbacks including slow tumor uptake, low tumor/normal tissues ratios, and non-specific metabolisms producing undesired metabolites [35]. Hence, the development of novel radiotracers with better physicochemical and biological properties, and the improvement of hypoxia imaging effectiveness are necessary for successful clinical applications in the future.

In the development of novel hypoxia radiotracers, several important criteria should be met to achieve high-quality PET or SPECT images, namely, high accumulation and retention at tumor sites yet low uptakes in blood and normal tissues. Many efforts have been made to obtain the probable pharmacokinetics of hypoxia radiotracers. Recently developed ^18^F-radiotracers for hypoxia can be categorized into two main groups: radiotracers with linkers and radiotracers with carbohydrate structures. For example, [^18^F]FMISO, and [^18^F]EF5, two commonly used radiotracers, are radiotracers with linkers. These types of tracers have a simple structure and a good hydrophilic property as well as a high uptake in the hypoxia region. Second, radiotracers with carbohydrate structures are also used in the PET study. [^18^F]FAZA, which is another commonly used radiotracer, is a radiotracer with a carbohydrate structure. These types of tracers showed a high hydrophilic property due to hydroxy groups and a high uptake in the hypoxia region. On the other hand, ^99m^Tc-radiotracers can be classified based on the number of nitroimidazole moieties present in their structures, specifically, mono-nitroimidazole, di-nitroimidazole, and multi-nitroimidazole. The main structural difference of recently developed ^99m^Tc-labeled radiotracers is the number of nitroimidazole and related linkers or chelates which affected the properties of the radiotracers. For example, the addition of more nitroimidazole can increase the accumulation of nitroimidazole-bearing radiotracers in the hypoxia region.

In recent studies, there has been a noticeable trend towards adjusting the hydrophilicity of radiotracers in order to achieve higher tumor uptake and tumor/background contrast compared to the common radiotracer [^18^F]FMISO. Notably, in many developed radiotracers, the superior radiotracer with the highest tumor uptake or the highest tumor/normal tissue ratios is often more properly hydrophilic than the other radiotracers in the group. In several studies, the effect of linkers on the hydrophilicity and biological properties of radiotracers has been investigated. The length of the linkers has revealed a great impact on the lipophilicity of radiotracers, resulting in notable changes in their uptake and retention in tumor and normal tissues. In this approach, PEG chains are commonly used to connect the nitroimidazole moieties to the rest of the tracer containing the radioisotope, thereby leading to the proper hydrophilicity of the radiotracers compared to those without a PEG chain. Similarly, extending the CH_2_ chains also makes the radiotracers more lipophilic. In addition, adding benzene moieties can also increase the lipophilicity of the radiotracer while adding more nitroimidazole moieties might affect the hydrophilicity of the radiotracers, depending on their overall structure. The advantage of highly hydrophilic radiotracers is that they have a fast clearance from blood; thus, their tumor/blood ratios were significantly increased, PET/SPECT image contrasts were greatly improved and the radiotracers were excreted via renal routes rapidly. However, increasing hydrophilicity is not always correlated with the proper pharmacokinetics and the best tumor/background contrast, as in the cases of [^99m^Tc]**30**, [^99m^Tc]**37**, and [^131^I]**85**. This might be explained that these radiotracers were cleared from blood too quickly that they did not have enough time to absorb into cancer cells, as well as not being lipophilic enough to enter cancer cells by diffusion through phospholipid bilayers. In contrast, highly lipophilic radiotracers can be retained in cancer and normal tissues. Additionally, their long retention in normal tissues significantly reduced the tumor/normal tissues ratios and PET/SPECT image contrast. Therefore, in the development of novel radiotracers for hypoxia, it is important to adjust the hydrophilicity of the radiotracers in order to find the radiotracer with optimal hydrophilicity and pharmacokinetics.

We believe that criteria other than hydrophilicity can also affect the pharmacokinetics of the radiotracers and should be considered when developing novel radiotracers for hypoxia. Adding multiple nitroimidazole moieties is also a commonly employed approach in the design of novel radiotracers to enhance better uptake in the hypoxia regions of tumors. Notably, multiple nitroimidazole units can be added into a radiolabeled complex bearing bifunctional chelators to capture radioactive transition metals like ^64^Cu or ^99m^Tc. For examples, three and up to six nitroimidazole units can be added into ^99m^Tc-labeled complexes, while ^18^F-labeled radiotracers have one or two nitroimidazole units. [^99m^Tc]**63** and [^99m^Tc]**66** are the two examples for which the presence of more nitroimidazole resulted in better contrast.

^18^F has been the most used radioisotope in the development of hypoxia radiotracers for many decades due to its small size and inert characteristics [183]. Most of the ^18^F radiotracers were prepared via nucleophilic substitution reactions. However, in order to prepare ^18^F radiotracers, big and expensive cyclotrons are required. Thus, a generator, which is a more simple and easy-to-handle piece of equipment to produce radionuclides such as ^99m^Tc, is also popular nowadays. It is clear that besides the most common radioisotope ^18^F, coordination of the ^99m^Tc core to a bifunctional chelate has received growing attention recently due to many reasons. First, the convenience of generators over cyclotrons has made the preparation of ^99m^Tc-labeled radiotracers easier for both research and clinical purposes. Secondly, ^99m^Tc-labeled radiotracers are highly versatile owing to the use of bifunctional chelates, which are diverse and extensively studied [184,185,186,187,188]. Thus, future research in developing novel ^99m^Tc-labeled radiotracers for hypoxia should consider employing a variety of bifunctional chelates and nitroimidazole moieties. Thirdly, hydrophilic ^99m^Tc-labeled radiotracers can be synthesized from the corresponding precursors containing several hydrophilic groups, whereas the radiofluorination of precursors bearing several hydrophilic groups (mostly via nucleophilic substitution) is more difficult. Moreover, ^99m^Tc cores are varied in oxidation states, for example, ^99m^Tc(I) ([^99m^Tc(CO)_3_]^+^ core), ^99m^Tc(III) (^99m^Tc^3+^ core), ^99m^Tc(V) ([^99m^TcN]^2+^ core, [^99m^TcO]^3+^ core), which showed different biodistributions. However, tumor uptake values and tumor/background ratios seem to depend on many factors rather than only the oxidation state of the ^99m^Tc core. For instance, both [^99m^Tc]**37** and [^99m^Tc]**70** contained [^99m^TcO]^3+^ cores but [^99m^Tc]**37** exhibited the highest tumor uptakes as well as tumor/background ratios when compared to other analogues containing [^99m^TcN]^2+^ or [^99m^Tc(CO)_3_]^+^ cores, while [^99m^Tc]**70** exhibited higher tumor uptake and tumor/muscle ratio but lower tumor/blood ratio than [^99m^Tc]**69** ([^99m^TcN]^2+^ core). Therefore, side-by-side studies are still needed to study the effect of ^99m^Tc cores on the biological properties of ^99m^Tc-radiotracers for hypoxia. The main limitation of ^99m^Tc-labeled radiotracers is that SPECT imaging offers a lower sensitivity and accuracy compared to PET imaging [189,190]. Nonetheless, ^99m^Tc-labeled radiotracers have less stable coordination than the covalent bonds of radioactive fluorine and iodine.

Despite the advantages and favorable physicochemical and biological properties of the summarized ^99m^Tc-labeled radiotracers, there is still a lack of clinical trials conducted for these radiotracers. However, some ^18^F-labeled radiotracers such as [^18^F]FMISO, [^18^F]FAZA, and [^18^F]EF5 have been used in clinical studies. Thus, we expect that ^18^F-labeled radiotracers with proper linkers, carbohydrates, and nucleic acid could be used for clinical study. Validating PET/SPECT imaging tracers for hypoxia poses several challenges. In addition to common requirements such as non-toxicity, high uptake, and rapid clearance, the radiotracers must exhibit a suitable biodistribution specific to the different tumor types. As a result, many radiotracers lack universality across various types of cancers [191,192]. However, after solving these issues, radiolabeled radiotracers can be successfully used for clinical study.

Moreover, among nitroimidazole moieties, 2-nitroimidazole is the most widely studied because of its higher reduction potentials (−380 mV to −400 mV) than 5-nitroimidazole (−440 mV to −460 mV) and 4-nitroimidazole (−421 mV to −450 mV) [193]. However, it is notable that 4-nitroimidazole and 5-nitroimidazole are also very promising.

In recent years, the synthesis and evaluation of ^131/124^I- and ^64^Cu-labeled radiotracers are still limited, which might be due to their long half-life and production using cyclotrons. Thus, further research into the imaging techniques and equipment is needed to develop improved radiotracers with these radioactive isotopes in the imaging of tumor hypoxia.

Each radiotracer mentioned in this review has its advantages and disadvantages in lipophilicity, hypoxic selectivity, tumor uptake, tumor/blood contrast, tumor/muscle contrast, etc. Thus, the discovery of better novel hypoxia-targeting agents as imaging agents for wide applications in clinical hypoxia imaging is still needed.

Moreover, in order to visualize hypoxia through molecular imaging, several studies must be achieved for better molecular imaging in the future. First, proper radiotracers with more specificity and selectivity to hypoxia should be developed for a variety of applications and should be applied to clinical study. Thus, future studies should focus on the production of novel promising structures. Second, the preparation process for radiotracers should be achieved via more simple and efficient steps. Particularly, a short, low-cost, and environmentally synthetic process is useful. Third, radiolabeling protocol should also be easier and more effective; thus, desired radiolabeled compounds should be obtained with high radiochemical yields. Fourth, improved imaging techniques and equipment should also be developed to visualize hypoxia more clearly. It is expected that many scientists will endeavor to synthesize and evaluate novel hypoxia radiotracers with better properties for clinical application in the future. We believe that this review provides an overall picture of recent developments in new radiotracers for hypoxia.

## Figures and Tables

**Figure 1 pharmaceutics-15-01840-f001:**
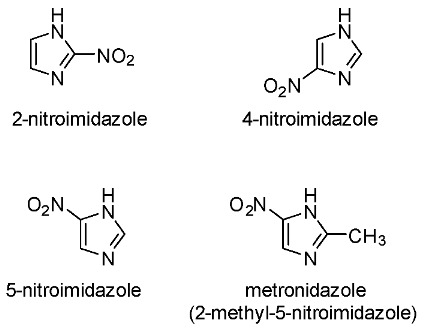
Chemical structures of nitroimidazole compounds.

**Figure 2 pharmaceutics-15-01840-f002:**
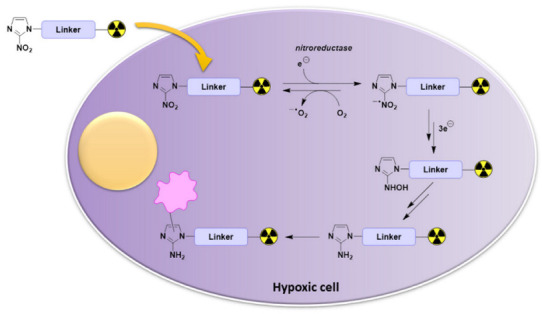
Oxygen-dependent bioreduction and retention of radiotracers containing nitroimidazole moieties in hypoxic cells.

**Figure 3 pharmaceutics-15-01840-f003:**
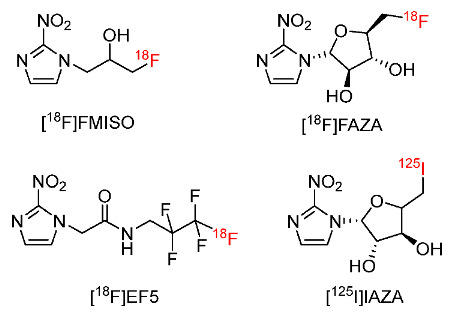
Chemical structures of commonly used hypoxia PET imaging agents containing nitroimidazole moieties.

**Figure 4 pharmaceutics-15-01840-f004:**
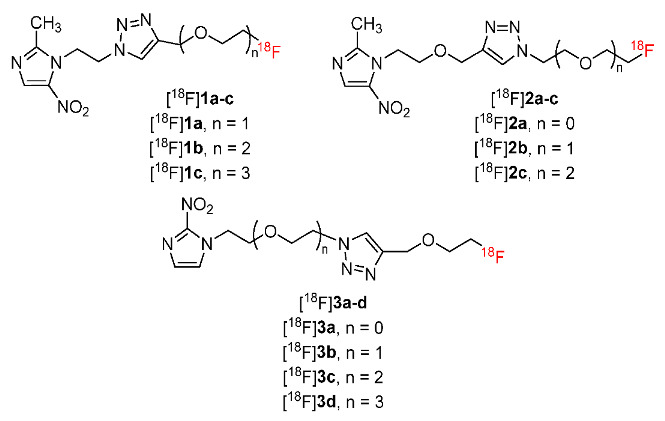
Chemical structures of ten ^18^F-labeled PEG-modified nitroimidazole derivatives.

**Figure 5 pharmaceutics-15-01840-f005:**
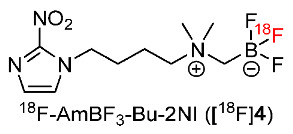
Chemical structure of ^18^F-AmBF_3_-Bu-2NI.

**Figure 6 pharmaceutics-15-01840-f006:**
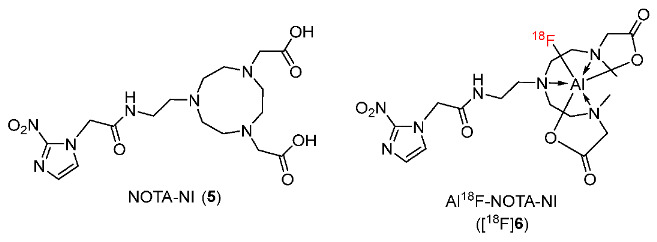
Chemical structures of precursor NOTA-NI and radiotracer Al^18^F-NOTA-NI.

**Figure 7 pharmaceutics-15-01840-f007:**
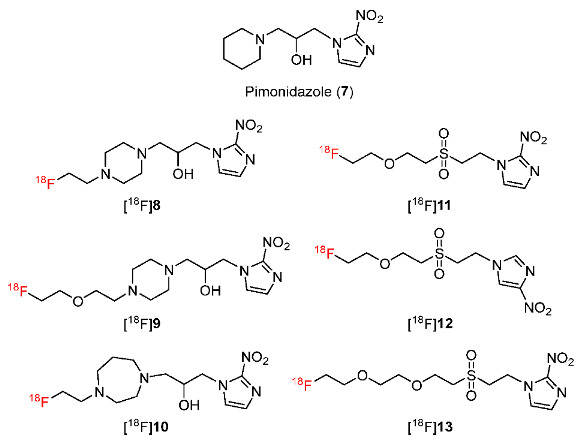
Chemical structures of ^18^F-labeled pimonidazole derivatives and nitroimidazole derivatives bearing sulfonyl linkers.

**Figure 8 pharmaceutics-15-01840-f008:**
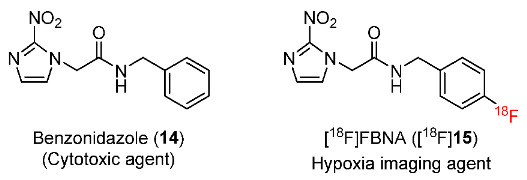
Chemical structures of benzonidazole and [^18^F]FBNA.

**Figure 9 pharmaceutics-15-01840-f009:**
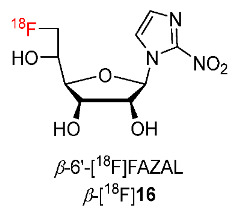
Chemical structure of *β*-6′-[^18^F]FAZAL (*β*-[^18^F]**16**).

**Figure 10 pharmaceutics-15-01840-f010:**
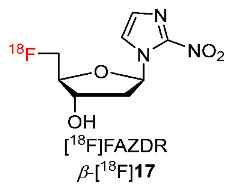
Chemical structure of [^18^F]FAZDR (*β*-[^18^F]**17**).

**Figure 11 pharmaceutics-15-01840-f011:**
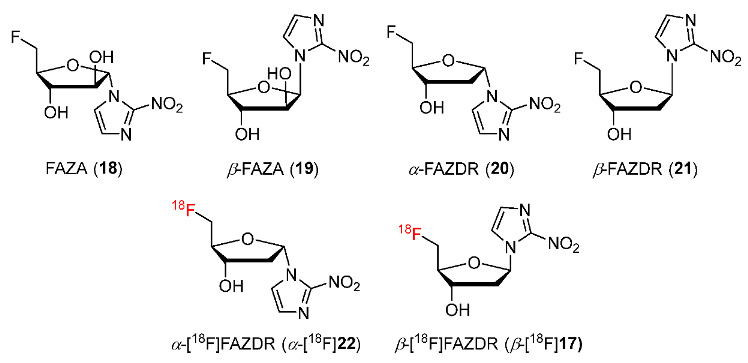
Chemical structures of precursors *β*-FAZA, FAZA, *α*-FAZDR and *β*-FAZDR, and radiotracers *α*-FAZDR and *β*-FAZDR.

**Figure 12 pharmaceutics-15-01840-f012:**
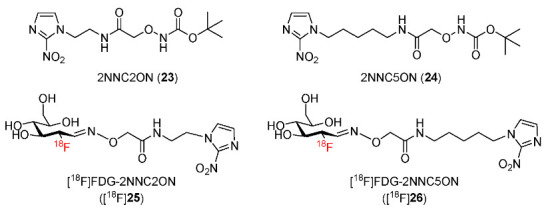
Chemical structures of aminooxy derivatives of 2-nitroimidazole and the corresponding [^18^F]FDG-labeled radiotracers.

**Figure 13 pharmaceutics-15-01840-f013:**
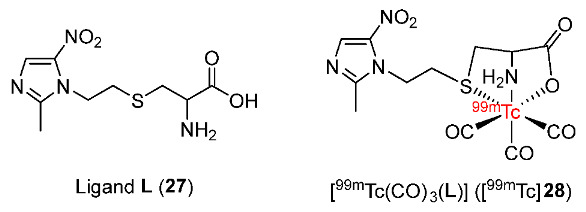
Chemical structures of ligand **L** and [^99m^Tc(CO)_3_(**L**)].

**Figure 14 pharmaceutics-15-01840-f014:**
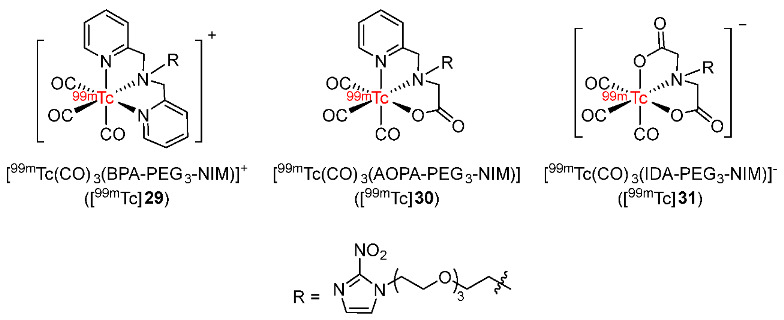
Chemical structures of [^99m^Tc(CO)_3_(BPA-PEG_3_-NIM)]^+^, [^99m^Tc(CO)_3_(AOPA-PEG_3_-NIM)] and ^99m^Tc(CO)_3_(IDA-PEG_3_-NIM)]^−^.

**Figure 15 pharmaceutics-15-01840-f015:**
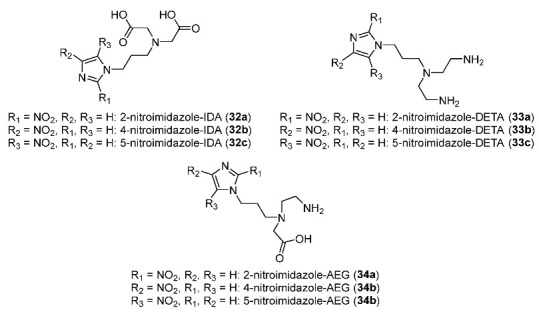
Chemical structures of tridentate ligands IDA, DETA and AEG containing 2-, 4- or 5-nitroimidazole moieties.

**Figure 16 pharmaceutics-15-01840-f016:**
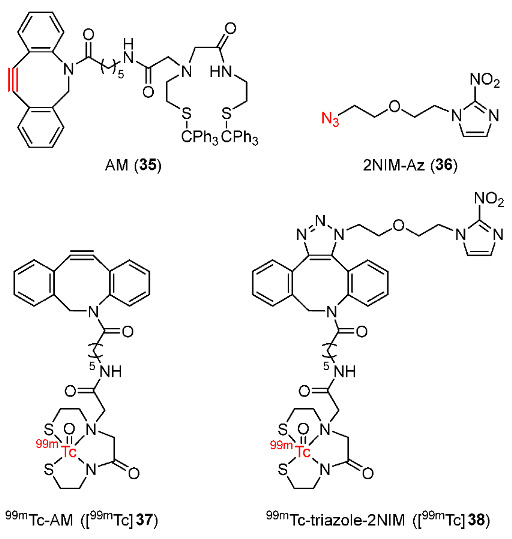
Chemical structures of azadibenzocyclooctyne-MAMA (AM), 2-nitroimidazole-azide (2NIM-Az), ^99m^Tc-AM and ^99m^Tc-triazole-2NIM.

**Figure 17 pharmaceutics-15-01840-f017:**
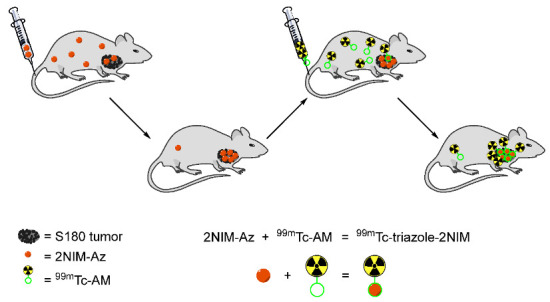
Illustration of a tumor hypoxia-pretargeting technique using an in vivo click reaction of 2NIM-Az and ^99m^Tc-AM.

**Figure 18 pharmaceutics-15-01840-f018:**
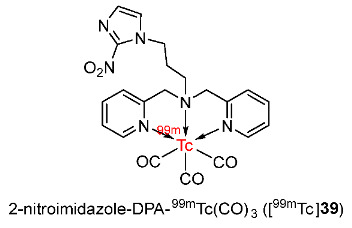
Chemical structure of 2-NI–DPA–^99m^Tc(CO)_3_ ([^99m^Tc]**39**).

**Figure 19 pharmaceutics-15-01840-f019:**
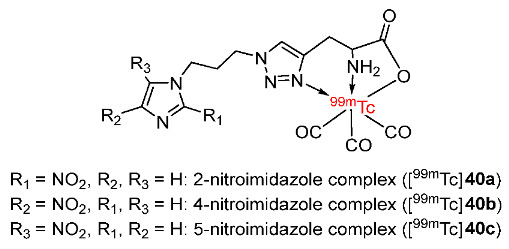
Chemical structures of ^99m^Tc(CO)_3_-labeled triazole derivatives of 2-, 4- and 5-nitroimidazoles.

**Figure 20 pharmaceutics-15-01840-f020:**
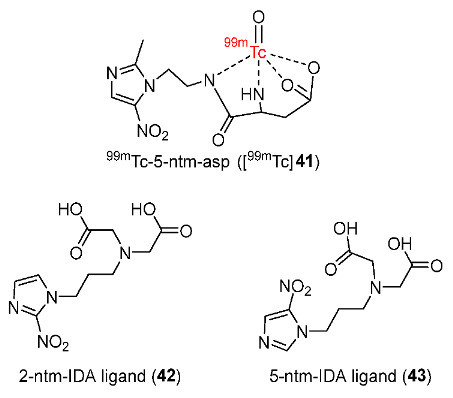
Chemical structures of ^99m^Tc-5-ntm-asp, 2-ntm-IDA, and 5-ntm-IDA ligands.

**Figure 21 pharmaceutics-15-01840-f021:**
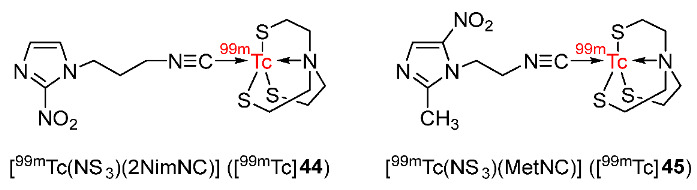
Chemical structures of [^99m^Tc(NS_3_)(2NimNC)] and [^99m^Tc(NS_3_)(MetNC)].

**Figure 22 pharmaceutics-15-01840-f022:**
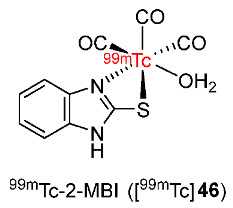
Chemical structure of ^99m^Tc-2-MBI complex.

**Figure 23 pharmaceutics-15-01840-f023:**
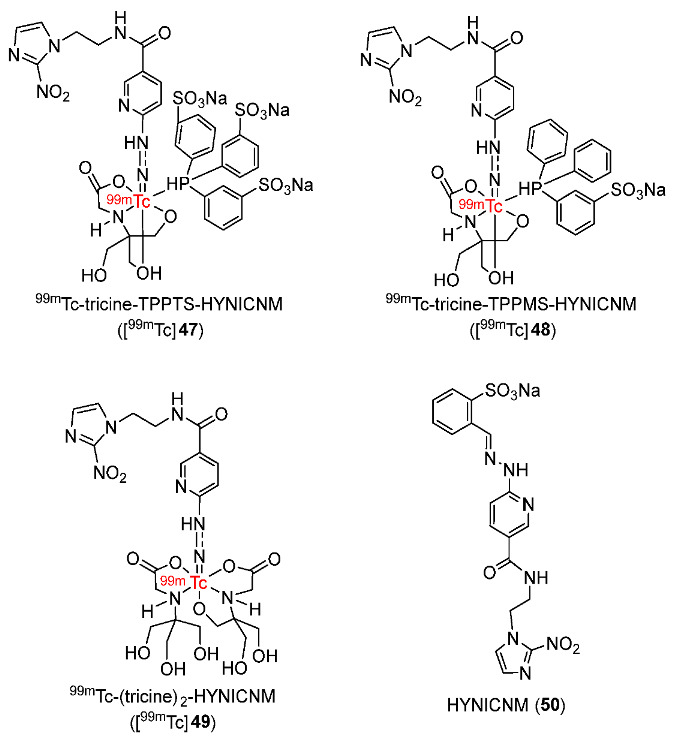
Chemical structures of ^99m^Tc-labeled complexes ^99m^Tc-tricine-TPPTS-HYNICNM, ^99m^Tc-tricine-TPPMS-HYNICNM, ^99m^Tc-(tricine)_2_-HYNICNM and HYNICNM ligand.

**Figure 24 pharmaceutics-15-01840-f024:**
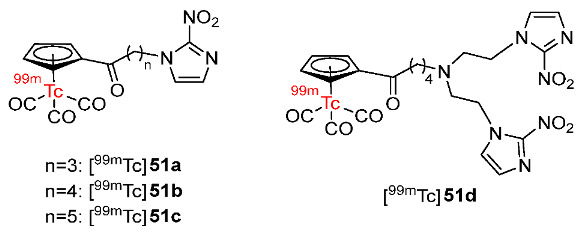
Chemical structures of cyclopentadienyl ^99m^Tc(CO)_3_ complexes containing 2-nitroimidazole.

**Figure 25 pharmaceutics-15-01840-f025:**
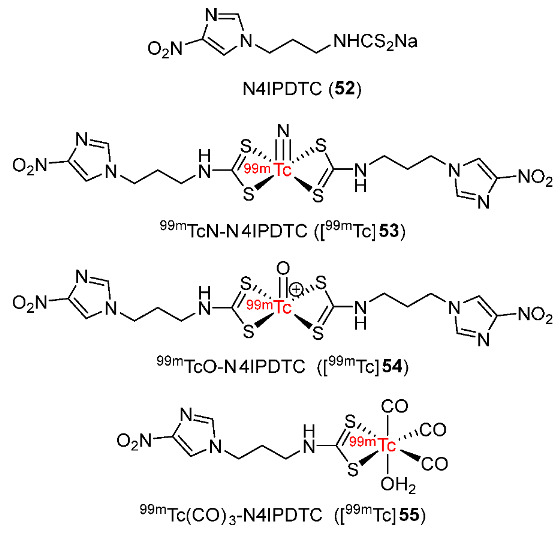
Chemical structures of ligand N4IPDTC and ^99m^TcN-N4IPDTC, ^99m^TcO-N4IPDTC and ^99m^Tc(CO)_3_-N4IPDTC.

**Figure 26 pharmaceutics-15-01840-f026:**
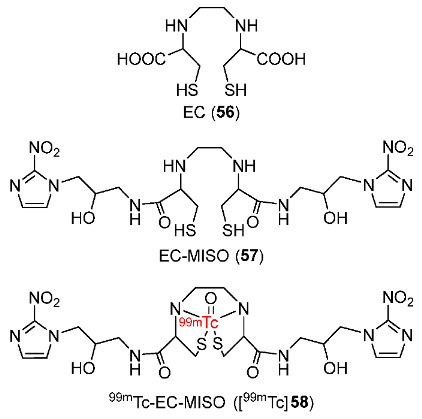
Chemical structures of precursor EC-MISO and ^99m^Tc-EC-MISO.

**Figure 27 pharmaceutics-15-01840-f027:**
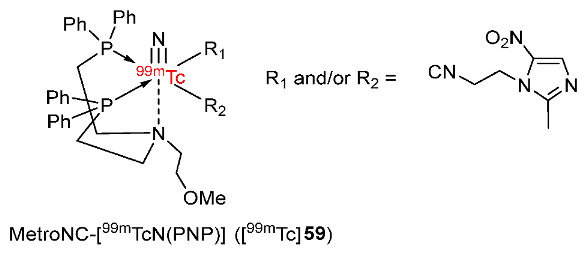
Chemical structures of MetroNC-[^99m^TcN(PNP)] complex.

**Figure 28 pharmaceutics-15-01840-f028:**
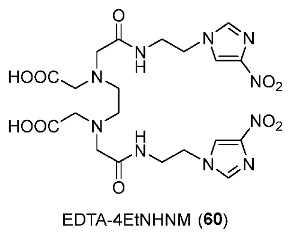
Chemical structure of EDTA-4-EtNHNM ligand.

**Figure 29 pharmaceutics-15-01840-f029:**
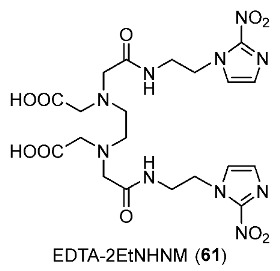
Chemical structure of EDTA-2-EtNHNM ligand.

**Figure 30 pharmaceutics-15-01840-f030:**
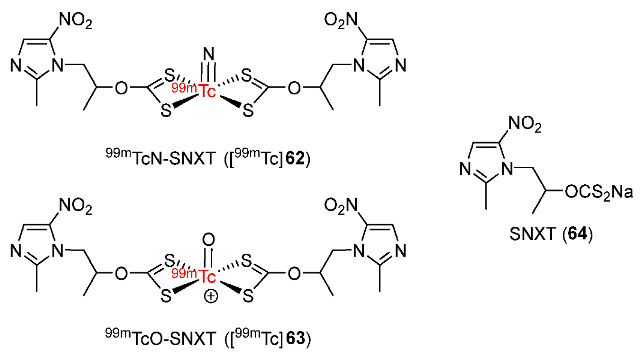
Chemical structures of ^99m^TcN-SNXT, ^99m^TcO-SNXT complexes, and SNXT ligand.

**Figure 31 pharmaceutics-15-01840-f031:**
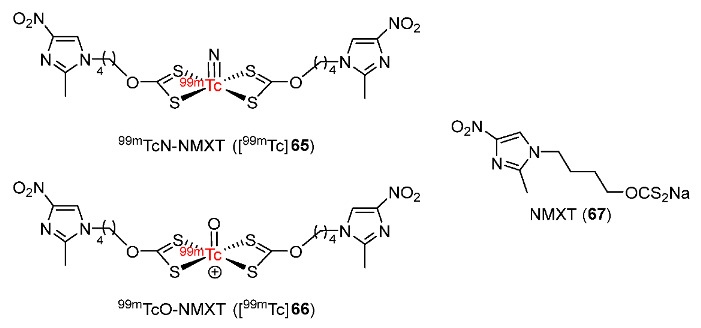
Chemical structures of ^99m^Tc-labeled complexes ^99m^TcN-NMXT, ^99m^TcO-NMXT and NMXT ligand.

**Figure 32 pharmaceutics-15-01840-f032:**
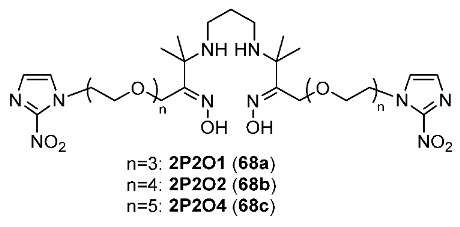
Chemical structures of 2P2O1, 2P2O2 and 2P2O4 ligands.

**Figure 33 pharmaceutics-15-01840-f033:**
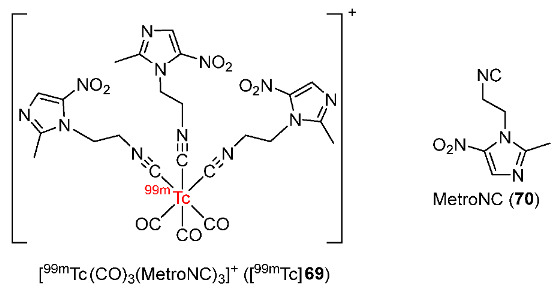
Chemical structures of [^99m^Tc(CO)_3_(MetroNC)_3_]^+^ complex and MetroNC ligand.

**Figure 34 pharmaceutics-15-01840-f034:**
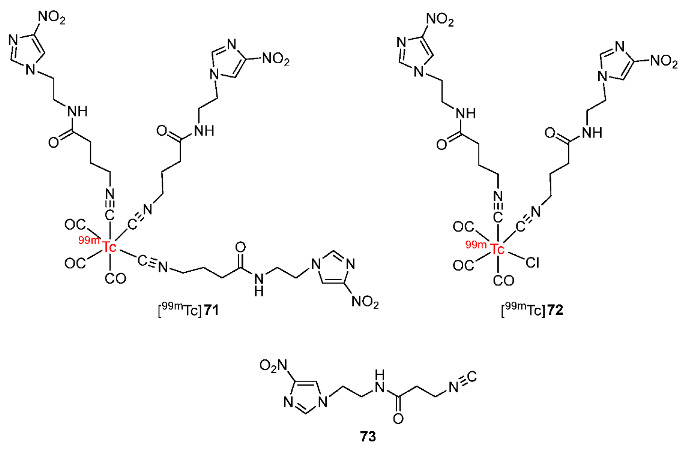
Chemical structures of [^99m^Tc]**71** and [^99m^Tc]**72** complexes.

**Figure 35 pharmaceutics-15-01840-f035:**
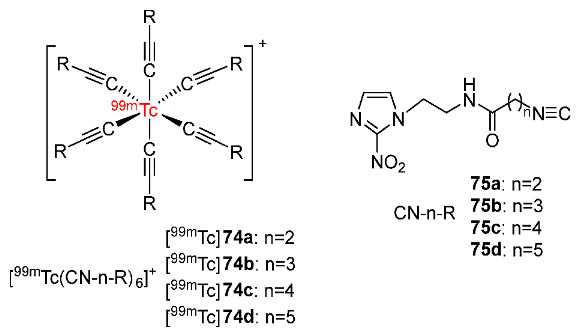
Chemical structures of complexes [^99m^Tc]**55a**–**d** and 2-nitroimidazole isocyanide ligands.

**Figure 36 pharmaceutics-15-01840-f036:**
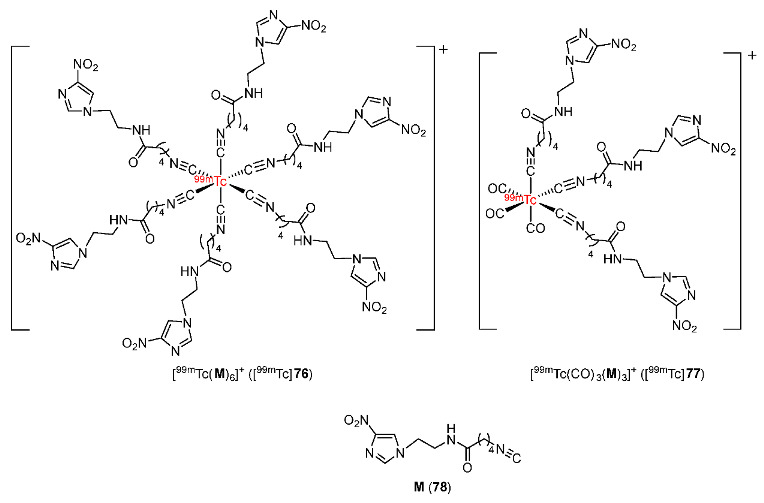
Chemical structures of [^99m^Tc(**M**)_6_]^+^, [^99m^Tc(CO)_3_(**M**)_3_]^+^ and ligand **M**.

**Figure 37 pharmaceutics-15-01840-f037:**
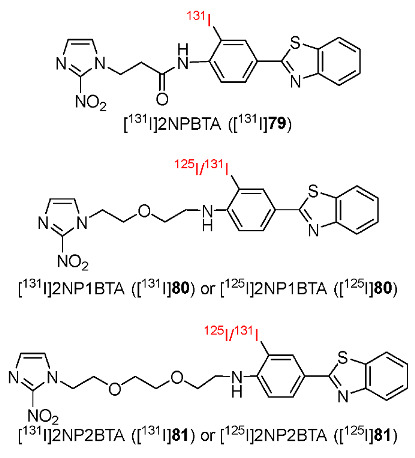
Chemical structures of previously reported radiotracer [^131^I]2NPBTA and ^131^I- and ^125^I-labeled radiotracers containing 2-nitroimidazole, PEG chain and BTA group.

**Figure 38 pharmaceutics-15-01840-f038:**
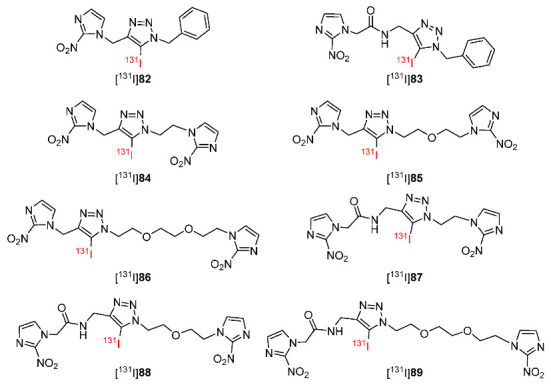
Chemical structures of ^131^I-radiolabeled 2-nitroimidazole derivatives.

**Figure 39 pharmaceutics-15-01840-f039:**
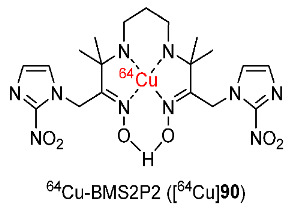
Chemical structure of ^64^Cu-BMS2P2.

## Data Availability

Not applicable.

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
