# Peer review of "Recent Developments in PET and SPECT Radiotracers as Radiopharmaceuticals for Hypoxia Tumors"

_pharmaceutics, 2023, doi:10.3390/pharmaceutics15071840_

Round 1

Reviewer 1 Report

The paper entitled “Recent developments in PET and SPECT radiotracers as radio-pharmaceuticals for hypoxia tumors” by Nguyen and Kim is an interesting review focusing on the recently developed radiopharmaceuticals for the detection of solid tumors characterized by hypoxia. The manuscript is well organized and the results from the cited papers are clearly presented. I also appreciated the presence of all the chemical structures of the radiopharmaceuticals discussed.

Moreover, English language is very fluent and easy to read.

I only have few minor suggestions to further improve the manuscript, which according to me is already a high-quality paper:

1)      Some references cited in the introduction are not very recent (for example, references 40-44, 56, 57, 60-62). I suggest the Authors to replace them with (or to add) some more recent paper. Among the Literature, I suggest the following papers:

On line 73, besides ref. 40-44, I suggest to add the following papers:

-          Ailuno, G. et al. The Pharmaceutical Technology Approach on Imaging Innovations from Italian Research. Pharmaceutics 2021, 13, 1214. https://doi.org/10.3390/pharmaceutics13081214

-          Kabasawa H. MR Imaging in the 21st Century: Technical Innovation over the First Two Decades. Magnetic Resonance in Medical Sciences 2022, 21. https://doi.org/10.2463/mrms.rev.2021-0011

-          Amirrashedi M. et al. Towards quantitative small-animal imaging on hybrid PET/CT and PET/MRI systems. Clinical and Translational Imaging 2020, 8, 4. DOI: 10.1007/s40336-020-00376-y

On line 80, besides ref. 56,57, I suggest to add the following papers:

-          Ćorović A. et al. Novel Positron Emission Tomography Tracers for Imaging Vascular Inflammation. Current Cardiology Reports 2020, 22. https://doi.org/10.1007/s11886-020-01372-4

-          Sammartino A.M. et al. “Vascular inflammation and cardiovascular disease: review about the role of PET imaging”.  The International Journal of Cardiovascular Imaging 2023, 39. https://doi.org/10.1007/s10554-022-02730-9

-          Pastorino S. et al. Two Novel PET Radiopharmaceuticals for Endothelial Vascular Cell Adhesion Molecule-1 (VCAM-1) Targeting. Pharmaceutics 2021, 13, 1025. https://doi.org/10.3390/pharmaceutics13071025

On line 81, besides ref. 60-62, I suggest to add the following papers:

-          Thientunyakit T. et al. Molecular PET Imaging in Alzheimer's Disease. Journal of Medical and Biological Engineering 2022, 42. https://doi.org/10.1007/s40846-022-00717-4

-          Mohamed M.A. et al. Astrogliosis in aging and Parkinson's disease dementia: a new clinical study with C-11-BU99008 PET. Brain Communications 2022. https://doi.org/10.1093/braincomms/fcac199

2)      I suggest the Authors to revise the “Conclusions” section. In particular, from line 875 to line 882, the Authors report some generic considerations about the features that the ideal hypoxia radiotracer should exhibit; I think this information are more suitable in the introduction, while in the conclusions I suggest the Authors to underline the importance of the progresses reached from some of the cited research groups, highlighting which strategies seem, according to the Authors’ opinion, the most promising.

3)      In the beginning of the subsections dedicated to the radiotracers labeled with the different radioisotopes, I suggest the Authors to briefly mention the advantages and the drawbacks of using each radiotracer.

Author Response

Thank you for considering our revised manuscript entitled “Recent developments in PET and SPECT radiotracers as radiopharmaceuticals for hypoxia tumors” for publication in Pharmaceutics.

We would like to start by thanking the reviewers for their constructive criticisms and encouraging remarks.

You will find below that we answered all the reviewers’ questions and followed their suggestions by adding more descriptions and explanations.

(Q1) For the comment, “1) Some references cited in the introduction are not very recent (for example, references 40-44, 56, 57, 60-62). I suggest the Authors to replace them with (or to add) some more recent paper. Among the Literature, I suggest the following papers:

On line 73, besides ref. 40-44, I suggest to add the following papers:

- Ailuno, G. et al. The Pharmaceutical Technology Approach on Imaging Innovations from Italian Research. Pharmaceutics 2021, 13, 1214. https://doi.org/10.3390/pharmaceutics13081214

- Kabasawa H. MR Imaging in the 21st Century: Technical Innovation over the First Two Decades. Magnetic Resonance in Medical Sciences 2022, 21. https://doi.org/10.2463/mrms.rev.2021-0011

- Amirrashedi M. et al. Towards quantitative small-animal imaging on hybrid PET/CT and PET/MRI systems. Clinical and Translational Imaging 2020, 8, 4. DOI: 10.1007/s40336-020-00376-y

On line 80, besides ref. 56,57, I suggest to add the following papers:

- Ćorović A. et al. Novel Positron Emission Tomography Tracers for Imaging Vascular Inflammation. Current Cardiology Reports 2020, 22. https://doi.org/10.1007/s11886-020-01372-4

- Sammartino A.M. et al. “Vascular inflammation and cardiovascular disease: review about the role of PET imaging”.  The International Journal of Cardiovascular Imaging 2023, 39. https://doi.org/10.1007/s10554-022-02730-9

- Pastorino S. et al. Two Novel PET Radiopharmaceuticals for Endothelial Vascular Cell Adhesion Molecule-1 (VCAM-1) Targeting. Pharmaceutics 2021, 13, 1025. https://doi.org/10.3390/pharmaceutics13071025

On line 81, besides ref. 60-62, I suggest to add the following papers:

- Thientunyakit T. et al. Molecular PET Imaging in Alzheimer's Disease. Journal of Medical and Biological Engineering 2022, 42. https://doi.org/10.1007/s40846-022-00717-4

- Mohamed M.A. et al. Astrogliosis in aging and Parkinson's disease dementia: a new clinical study with C-11-BU99008 PET. Brain Communications 2022. https://doi.org/10.1093/braincomms/fcac199”,

(A1) According to the comments, new references (Pharmaceutics 2021, 13, 1214; Magnetic Resonance in Medical Sciences 2022, 21, 71; Clinical and Translational Imaging 2020, 8, 4; Current Cardiology Reports 2020, 22, 119; The International Journal of Cardiovascular Imaging 2023, 39, 433; Pharmaceutics 2021, 13, 1025; Journal of Medical and Biological Engineering 2022, 42, 1; Brain Communications 2022, 42, fcac199) were added to the suggested parts of the manuscript. In additions, some recent papers (ref 37-55, and ref 59-68) were added into introduction section.

(Q2) For the comment, “2) I suggest the Authors to revise the “Conclusions” section. In particular, from line 875 to line 882, the Authors report some generic considerations about the features that the ideal hypoxia radiotracer should exhibit; I think this information are more suitable in the introduction, while in the conclusions I suggest the Authors to underline the importance of the progresses reached from some of the cited research groups, highlighting which strategies seem, according to the Authors’ opinion, the most promising. ”,

(A2-1) The following sentences from the conclusion section were moved into the Introduction section in page 4:

“…An ideal hypoxia radiotracer should exhibit several physicochemical and biological properties. For example, radiotracers for hypoxia should have high selectivity toward hypoxia with low retention in normal tissues and high retention in tumor sites. They should be non-toxic, easy to prepare, convenient [35], and have probable lipophilicity. Aside from hypoxic tissues, the degradation of radiotracers in normal tissues should only generate non-specific metabolites which cannot be trapped in these tissues [35]. Moreover, the trade-off between the absolute tumor uptake signal and the relative tumor/background ratio is also a concern [29]…”.

(A-2) The main progress of cited research groups were added into the Conclusion and Perspectives section in page 27 as the following paragraphs:

“…In recent studies, there has been a noticeable trend towards adjusting the hydrophilicity of radiotracers in order to achieve higher tumor uptake and tumor/background contrast compared to the common radiotracer [18F]FMISO. Notably, in many developed radiotracers, the superior radiotracer with the highest tumor uptake or the highest tumor/normal tissue ratios is often more proper hydrophilic than the other radiotracers in the group. In several studies, the effect of linkers on the hydrophilicity and biological properties of radiotracers has been investigated. The length of the linkers has shown a great impact on the lipophilicity of radiotracers, resulting in notable changes in their uptake and retention in tumor and normal tissues. In this approach, PEG chains are commonly used to connect the nitroimidazole moieties to the rest of the tracer containing the radioisotope, thereby making proper hydrophilic of the radiotracers compared to those without a PEG chain. Similarly, extending the CH2 chains also makes the radiotracers more lipophilic. In addition, adding benzene moieties can also increase the lipophilicity of the radiotracer while adding more nitroimidazole moieties might affect the hydrophilicity of the radiotracers, depending on their overall structure. The advantage of highly hydrophilic radiotracers is that they had fast clearance from blood, thus, the tumor/blood ratios were significantly increased, PET/SPECT image contrasts were greatly improved and the radiotracers were excreted via renal routes rapidly. However, increasing hydrophilicity is not always correlated with the proper pharmacokinetics and the best tumor/background contrast, as in the cases of [99mTc]30, [99mTc]37, and [131I]85. This might be explained that these radiotracers were cleared from blood too quickly that they did not have enough time to absorb into cancer cells, as well as they were not lipophilic enough to enter cancer cells by diffusion through phospholipid bilayers. In contrast, highly lipophilic radiotracers can retain in cancer and normal tissues. Additionally, their long retention in normal tissues significantly reduced the tumor/normal tissues ratios and PET/SPECT image contrast. Therefore, in the development of novel radiotracers for hypoxia, it is important to adjust the hydrophilicity of the radiotracers in order to find the radiotracer with optimal hydrophilicity and pharmacokinetics. 

We believe that criteria other than hydrophilicity can also affect the pharmacokinetics of the radiotracers and should be considered when developing novel radiotracers for hypoxia. Adding multiple nitroimidazole moieties is also a commonly employed approach in the design of novel radiotracers to enhance better uptake in hypoxia regions of tumors. Notably, multiple nitroimidazole units can be added into a radiolabeled complex bearing bifunctional chelators to capture radioactive transition metals like 64Cu or 99mTc. For examples, three and up to six nitroimidazole units can be added into 99mTc-labeled complexes, while 18F-labeled radiotracers have one or two nitroimidazole units. [99mTc]63 and [99mTc]66 are the two examples that the presence of more nitroimidazole resulted in better contrast…”

(Q3) For the comment, “3) In the beginning of the subsections dedicated to the radiotracers labeled with the different radioisotopes, I suggest the Authors to briefly mention the advantages and the drawbacks of using each radiotracer.”,

(A3) The following paragraph was added into the beginning of the subsection “2.1. 18F-Radiotracers for hypoxia” (page 5) to mention the advantages and disadvantages of 18F:

18F is a positron-emitting radioisotope with a half-life of 110 min. Up to now, 18F is still the most widely used radioisotope for the preparation of hypoxia-targeting radiopharmaceuticals due to its proper half-life allowing extending PET scans and distribution to distant facilities, as well as low positron energy (0.635 MeV), high electron intensity and high resolution [123,124]. Moreover, 18F is small in size and chemically inert, allowing it to easily incorporate into the structures of radiotracers without greatly affecting the physicochemical and biological properties [125,126]. However, the production of 18F requires a cyclotron which is high-cost and takes up large space [127], Since the development of [18F]FMISO, the first radiotracer for imaging of hypoxia, various 18F-labeled analogues of 2-nitroimidazole for hypoxia have been extensively studied both preclinically and clinically [128,129]. …”.

The following paragraph was added into the beginning of the subsection “2.2. 99mTc-Radiotracers for hypoxia” (page 10) to mention the advantages and disadvantages of 99mTc:

99mTc is a radionuclide emitting gamma radiation widely used for SPECT imaging. 99mTc possesses a favorable half-life of 6 hours and low photon energy of 140 keV. In terms of convenience, compared to 18F, 99mTc can be obtained on-site as a pertechnetate (99mTcO4) by using commercial 99Mo/99mTc generators which are smaller and more affordable than cyclotrons [139]. The preparation of 99mTc-labeled complexes through coordination reactions is often conducted smoothly with high yields. However, the obtained radiolabeled products might exhibit physical and biological properties distinctly different from their precursors due to chelates. Moreover, degradation and transchelation of the 99mTc-labeled complexes should be noticed because these factors might affect the stability and radiopharmaceutical applications of the complexes [140]…”.

The following paragraph was added into the beginning of the subsection “2.3. 131I-/125I-Radiotracers for hypoxia” (page 24) to mention the advantages and disadvantages of radioactive iodine:

“Similar to 18F, radioactive iodine can also be directly incorporated into various molecules but retain their biological properties. Radionuclide 124I has a long half-life of 4 days which enables the distant distribution as well as long-term PET imaging studies [172,173]. 131I emits beta irradiation and is used for the preparation of SPECT imaging agents. 125I, a SPECT radionuclide emitting photon irradiation, has a much longer half-life than 124I (59.5 days) [174]. This long half-life of 125I might not be favorable for in vivo SPECT imaging due to the long exposure of the body to radiation. As a radioactive halogen like 18F, the yields of radioiodination reactions might not always be high, limiting the application of radioactive iodine. Additionally, iodine radionuclides are produced by using a cyclotron, which is another drawback…”.

The following paragraph was added into the beginning of the subsection “2.4. 64Cu-Radiotracers for hypoxia” (page 25) to mention the advantages and disadvantages of 64Cu.

64Cu is one of the copper radioisotopes used in molecular imaging and radiotherapy. With a half-life of 12.7 hours, many 64Cu-labeled radiotracers were developed for the PET imaging of many types of cancers [178]. The production of 64Cu-labeled radiotracers is mainly via the coordination of a variety of chelators to 64Cu, which can be obtained from a cyclotron or reactor. Similar to 99mTc, the coordination of 64Cu might be unstable, which will affect the stability of the radiotracer in vivo. In addition, the inconvenience of using a cyclotron to generate 64Cu is also another drawback of 64Cu-labeled radiotracers…”.

We hope that our modifications to the manuscript for the specific concerns and questionable points will satisfy the reviewers and the requirements for the publication of this manuscript.

Reviewer 2 Report

The provided review paper by Nguyen and Kim is well-structured and summarizes relevent papers in the field of hypoxia targetting. The authors focused on the published papers of radiolabelled nitroimidazoles as bioreducible compounds.

The radiotracers are grouped by the type of the radionuclide and relevant data for their biodistribution and lypophilicity are provided. Nevertheless, the collected information is difficult to follow as it comments on almost 90 compounds.

The final conclusion is not a conclusion - it merely repeats part of the abstract and general information. The discussed radionuclides have very different chenistry - from halogenides to redox-active metal ions. This might as well have some contribution to the biochemical properties of the compounds discussed. 

In order for the review paper to be useful to the reader it should reflect the authors vision on the general trends in the advantages and disadvantages of all discussed structures. Group them as "more" promissing classes or less applicable ones; highlight the disadvantages and advantages in their discussed properties and observed tendencies related to their structures and/or stability. And finally, draw true perspectives on what is relevant for clinical trials by also mentioning or commenting on eventual data from clinical trials.

My final suggestion is to accept the paper after major revision in which the conclusion must be rewritten. Try to provide also data for clinical trials and to group the 99mTc-labelled compounds by some common characteristic and possibly discuss them as groups with common features (Tc(CO)cores, charged vs non-charged, etc.)

Author Response

Thank you for considering our revised manuscript entitled “Recent developments in PET and SPECT radiotracers as radiopharmaceuticals for hypoxia tumors” for publication in Pharmaceutics.

We would like to start by thanking the reviewers for their constructive criticisms and encouraging remarks.

You will find below that we answered all the reviewers’ questions and followed their suggestions by adding more descriptions and explanations.

(Q1) For the comment, “1. " The final conclusion is not a conclusion - it merely repeats part of the abstract and general information. The discussed radionuclides have very different chenistry - from halogenides to redox-active metal ions. This might as well have some contribution to the biochemical properties of the compounds discussed....." ”,

(A1) The following paragraphs were added into the beginning of each subsection to provide information of each radioisotope which might affect the properties of the radiotracers.

18F is a positron-emitting radioisotope with a half-life of 110 min. Up to now, 18F is still the most widely used radioisotope for the preparation of hypoxia-targeting radiopharmaceuticals due to its proper half-life allowing extending PET scans and distribution to distant facilities, as well as low positron energy (0.635 MeV), high electron intensity and high resolution [123,124]. Moreover, 18F is small in size and chemically inert, allowing it to easily incorporate into the structures of radiotracers without greatly affecting the physicochemical and biological properties [125,126]..”.

99mTc is a radionuclide emitting gamma radiation widely used for SPECT imaging. 99mTc possesses a favorable half-life of 6 hours and low photon energy of 140 keV. In terms of convenience, compared to 18F, 99mTc can be obtained on-site as a pertechnetate (99mTcO4) by using commercial 99Mo/99mTc generators which are smaller and more affordable than cyclotrons [139]. The preparation of 99mTc-labeled complexes through coordination reactions is often conducted smoothly with high yields. However, the obtained radiolabeled products might exhibit physical and biological properties distinctly different from their precursors due to chelates. Moreover, degradation and transchelation of the 99mTc-labeled complexes should be noticed because these factors might affect the stability and radiopharmaceutical applications of the complexes [140]…”. .

“Similar to 18F, radioactive iodine can also be directly incorporated into various molecules but retain their biological properties. Radionuclide 124I has a long half-life of 4 days which enables the distant distribution as well as long-term PET imaging studies [172,173]. 131I emits beta irradiation and is used for the preparation of SPECT imaging agents. 125I, a SPECT radionuclide emitting photon irradiation, has a much longer half-life than 124I (59.5 days) [174]. This long half-life of 125I might not be favorable for in vivo SPECT imaging due to the long exposure of the body to radiation. As a radioactive halogen like 18F, the yields of radioiodination reactions might not always be high, limiting the application of radioactive iodine. …”.

64Cu is one of the copper radioisotopes used in molecular imaging and radiotherapy. With a half-life of 12.7 hours, many 64Cu-labeled radiotracers were developed for the PET imaging of many types of cancers [178]. The production of 64Cu-labeled radiotracers is mainly via the coordination of a variety of chelators to 64Cu, which can be obtained from a cyclotron or reactor. Similar to 99mTc, the coordination of 64Cu might be unstable, which will affect the stability of the radiotracer in vivo…”.

(Q2) For the comment, “2. " In order for the review paper to be useful to the reader it should reflect the authors vision on the general trends in the advantages and disadvantages of all discussed structures. ..." ”,

(A2) Two general trends were added into the Conclusion and Perspectives section in page 27 as the following paragraphs:

“…In recent studies, there has been a noticeable trend towards adjusting the hydrophilicity of radiotracers in order to achieve higher tumor uptake and tumor/background contrast compared to the common radiotracer [18F]FMISO. Notably, in many developed radiotracers, the superior radiotracer with the highest tumor uptake or the highest tumor/normal tissue ratios is often more proper hydrophilic than the other radiotracers in the group. In several studies, the effect of linkers on the hydrophilicity and biological properties of radiotracers has been investigated. The length of the linkers has shown a great impact on the lipophilicity of radiotracers, resulting in notable changes in their uptake and retention in tumor and normal tissues. In this approach, PEG chains are commonly used to connect the nitroimidazole moieties to the rest of the tracer containing the radioisotope, thereby making proper hydrophilic of the radiotracers compared to those without a PEG chain. Similarly, extending the CH2 chains also makes the radiotracers more lipophilic. In addition, adding benzene moieties can also increase the lipophilicity of the radiotracer while adding more nitroimidazole moieties might affect the hydrophilicity of the radiotracers, depending on their overall structure. The advantage of highly hydrophilic radiotracers is that they had fast clearance from blood, thus, the tumor/blood ratios were significantly increased, PET/SPECT image contrasts were greatly improved and the radiotracers were excreted via renal routes rapidly. However, increasing hydrophilicity is not always correlated with the proper pharmacokinetics and the best tumor/background contrast, as in the cases of [99mTc]30, [99mTc]37, and [131I]85. This might be explained that these radiotracers were cleared from blood too quickly that they did not have enough time to absorb into cancer cells, as well as they were not lipophilic enough to enter cancer cells by diffusion through phospholipid bilayers. In contrast, highly lipophilic radiotracers can retain in cancer and normal tissues. Additionally, their long retention in normal tissues significantly reduced the tumor/normal tissues ratios and PET/SPECT image contrast. Therefore, in the development of novel radiotracers for hypoxia, it is important to adjust the hydrophilicity of the radiotracers in order to find the radiotracer with optimal hydrophilicity and pharmacokinetics. 

We believe that criteria other than hydrophilicity can also affect the pharmacokinetics of the radiotracers and should be considered when developing novel radiotracers for hypoxia. Adding multiple nitroimidazole moieties is also a commonly employed approach in the design of novel radiotracers to enhance better uptake in hypoxia regions of tumors. Notably, multiple nitroimidazole units can be added into a radiolabeled complex bearing bifunctional chelators to capture radioactive transition metals like 64Cu or 99mTc. For examples, three and up to six nitroimidazole units can be added into 99mTc-labeled complexes, while 18F-labeled radiotracers have one or two nitroimidazole units. [99mTc]63 and [99mTc]66 are the two examples that the presence of more nitroimidazole resulted in better contrast…”

(Q3) For the comment, “3. "Group them as "more" promissing classes or less applicable ones”,

(A3) 18F-radiotracers for hypoxia was categorized into two main groups: radiotracers with linkers and radiotracers with carbohydrate structures.

In the text (2. Development of Radiotracers for Hypoxia), the 18F-labeled radiotracers are divided into two sub-sections as following:

2.1.1. 18F-Radiotracers with linkers for hypoxia

2.1.2. 18F-Radiotracers with carbohydrate structure for hypoxia

And the following paragraphs were added into the Conclusion and Perspectives section in pages 27 and 28 to provide the description about grouping of 18F-radiotracers.

“…  Recently developed 18F-radiotracers for hypoxia can be categorized into two main groups: radiotracers with linkers and radiotracers with carbohydrate structures. First, radiotracers with linkers are widely used in PET study for hypoxia. For example, [18F]FMISO, and [18F]EF5, commonly used radiotracers, are radiotracers with linkers. These types of tracers have simple structure and good hydrophilic property as well as high uptake in hypoxia region. Second, radiotracers with carbohydrate structures are also used in PET study. [18F]FAZA, which is another commonly used radiotracer, is radiotracers with carbohydrate structures. These types of tracers showed high hydrophilic property due to hydroxy groups and high uptake in hypoxia region.

Besides, 99mTc-radiotracers for hypoxia was categorized into three main groups: radiotracers with mono-nitroimidazole, di-nitroimidazole, and multi-nitroimidazole.

In the text (2. Development of Radiotracers for Hypoxia), the 99mTc-labeled radiotracers were divided into three groups according to the number of nitroimidazole units in the radiotracers as following:

2.2.1. 99mTc-Radiotracers with mono-nitroimidazole for hypoxia”

2.2.2. 99mTc-Radiotracers with di-nitroimidazole for hypoxia”

2.2.3. 99mTc-Radiotracers with multi-nitroimidazole for hypoxia

And the following paragraphs were added into the Conclusion and Perspectives section in pages 27 and 28 to provide the description about grouping of 99mTc-radiotracers.

“… On the other hand, 99mTc-radiotracers can be classified based on the number of nitroimidazole moieties present in their structures, specifically mono-nitroimidazole, di-nitroimidazole, and multi-nitroimidazole. The main structural difference of recently developed 99mTc-labeled radiotracers is number of nitroimidazole and related linkers or chelates which affected properties of radiotracers. For examples, addition of more nitroimidazole can increase accumulation in hypoxia region…”

(Q4) For the comment, “4. "draw true perspectives on what is relevant for clinical trials by also mentioning or commenting on eventual data from clinical trials”,

(A4) The following paragraphs were added into the Conclusion and Perspectives section in pages 26 and 27 to provide our perspectives on what is relevant for clinical trials:

We checked all references for this review, however, there is no data for clinical trials

This review covered the development of radiopharmaceuticals radiolabeled with various radioisotopes for PET/SPECT studies of tumor hypoxia between 2014 and the beginning of 2023. Thus, all radiotracers were studies for animal study, before entering clinical trials.

However, the following sentences is added to comments.

“…Despite the advantages and favorable physicochemical and biological properties of the summarized 99mTc-labeled radiotracers, there is still a lack of clinical trials conducted for these radiotracers. However, some 18F-labeled radiotracers such as [18F]FMISO,  [18F]FAZA, [18F]EF5 have been for clinical study. Thus we expect that 18F-labeled radiotracers have proper linkers, carbohydrates, and nucleic acid could be used for clinical study. Validating PET/SPECT imaging tracers for hypoxia poses several challenges. In addition to common requirements such as non-toxicity, high uptake, and rapid clearance, the radiotracers must exhibit suitable biodistribution specific to different tumor types. As a result, many radiotracers lack universality across various types of cancers [191, 192]. However, after solving these issues, radiolabeled radiotracers can be successfully used for clinical study…”

(Q5) For the comment, “5. " Try to provide also data for clinical trials and group the 99mTc-labelled compounds by some common characteristic...." ”,

(A5-1) We checked all references for this review, however, there is no data for clinical trials

This review covered the development of radiopharmaceuticals radiolabeled with various radioisotopes for PET/SPECT studies of tumor hypoxia between 2014 and the beginning of 2023. Thus, all radiotracers were studies for animal study, before entering clinical trials.

(A5-2) 18F-radiotracers for hypoxia was categorized into two main groups: radiotracers with linkers and radiotracers with carbohydrate structures.

In the text (2. Development of Radiotracers for Hypoxia), the 18F-labeled radiotracers are divided into two sub-sections as following:

2.1.1. 18F-Radiotracers with linkers for hypoxia

2.1.2. 18F-Radiotracers with carbohydrate structure for hypoxia

And the following paragraphs were added into the Conclusion and Perspectives section in pages 27 and 28 to provide the description about grouping of 18F-radiotracers.

“…  Recently developed 18F-radiotracers for hypoxia can be categorized into two main groups: radiotracers with linkers and radiotracers with carbohydrate structures. First, radiotracers with linkers are widely used in PET study for hypoxia. For example, [18F]FMISO, and [18F]EF5, commonly used radiotracers, are radiotracers with linkers. These types of tracers have simple structure and good hydrophilic property as well as high uptake in hypoxia region. Second, radiotracers with carbohydrate structures are also used in PET study. [18F]FAZA, which is another commonly used radiotracer, is radiotracers with carbohydrate structures. These types of tracers showed high hydrophilic property due to hydroxy groups and high uptake in hypoxia region.

Besides, 99mTc-radiotracers for hypoxia was categorized into three main groups: radiotracers with mono-nitroimidazole, di-nitroimidazole, and multi-nitroimidazole.

In the text (2. Development of Radiotracers for Hypoxia), the 99mTc-labeled radiotracers were divided into three groups according to the number of nitroimidazole units in the radiotracers as following:

2.2.1. 99mTc-Radiotracers with mono-nitroimidazole for hypoxia”

2.2.2. 99mTc-Radiotracers with di-nitroimidazole for hypoxia”

2.2.3. 99mTc-Radiotracers with multi-nitroimidazole for hypoxia

And the following paragraphs were added into the Conclusion and Perspectives section in pages 27 and 28 to provide the description about grouping of 99mTc-radiotracers.

“… On the other hand, 99mTc-radiotracers can be classified based on the number of nitroimidazole moieties present in their structures, specifically mono-nitroimidazole, di-nitroimidazole, and multi-nitroimidazole. The main structural difference of recently developed 99mTc-labeled radiotracers is number of nitroimidazole and related linkers or chelates which affected properties of radiotracers. For examples, addition of more nitroimidazole can increase accumulation in hypoxia region…”

(Q6) For the comment, “3. "..possibly discuss them as groups with common features (Tc(CO)cores, charged vs non-charged, etc.)...." ”,

(A6) The following paragraphs were added into the Conclusion and Perspectives section in pages 27 and 28 to provide discussion about common feature of 99mTc cores.

“…It is clear that besides the most common radioisotope 18F, coordination of 99mTc core to a bifunctional chelate has received growing attention recently due to many reasons. First, the convenience of generators over cyclotrons has made the preparation of 99mTc-labeled radiotracers easier for both research and clinical purposes. Secondly, 99mTc-labeled radiotracers are highly versatile owing to the use of bifunctional chelates, which are diverse and extensively studied [183–187]. Thus, future research in developing novel 99mTc-labeled radiotracers for hypoxia should consider employing a variety of bifunctional chelates and nitroimidazole moieties. Thirdly, hydrophilic 99mTc-labeled radiotracers can be synthesized from the corresponding precursors containing several hydrophilic groups, whereas radiofluorination of precursors bearing several hydrophilic groups (mostly via nucleophilic substitution) is more difficult. Moreover, 99mTc cores are varied in oxidation states, for example, 99mTc(I) ([99mTc(CO)3]+ core), 99mTc(III) (99mTc3+ core), 99mTc(V) ([99mTcN]2+ core, [99mTcO]3+ core), which showed different biodistribution. However, tumor uptake values and tumor/background ratios seem to depend on many factors rather than only the oxidation state of the 99mTc core. For instance, both [99mTc]37 and [99mTc]70 contained [99mTcO]3+ cores but [99mTc]37 exhibited the highest tumor uptakes as well as tumor/background ratios when compared to other analogues containing [99mTcN]2+ or [99mTc(CO)3]+ cores, while [99mTc]70 exhibited higher tumor uptake and tumor/muscle ratio but lower tumor/blood ratio than [99mTc]69 ([99mTcN]2+ core). Therefore, side-by-side studies are still needed to study the effect of 99mTc cores on the biological properties of 99mTc-radiotracers for hypoxia. The main limitation of 99mTc-labeled radiotracers is that SPECT imaging offers lower sensitivity and accuracy compared to PET imaging [188,189]. Besides, 99mTc-labeled radiotracers have less stable coordination than the covalent bonds of radioactive fluorine and iodine.…”.

We hope that our modifications to the manuscript for the specific concerns and questionable points will satisfy the reviewers and the requirements for the publication of this manuscript.

Reviewer 3 Report

In this manuscript, the authors provide a detailed summary of different radionuclide labeled hypoxia markers that have been developed in the past decade, some of which are very interesting. The article also discusses problems associated with the hypoxia marker development. The reviewer recommends this article be published if the following comments are adequately addressed.

1)     In vivo and in vitro in the article should be corrected as in vivo and in vitro.

2)     Page 8, line 290, [18F]17exhibited should be [18F]17 exhibited.

3)     In the conclusion section, if the authors could forecast future directions of molecular imaging of hypoxia, it would be helpful to the readers.

It is OK

Author Response

Thank you for considering our revised manuscript entitled “Recent developments in PET and SPECT radiotracers as radiopharmaceuticals for hypoxia tumors” for publication in Pharmaceutics.

We would like to start by thanking the reviewers for their constructive criticisms and encouraging remarks.

You will find below that we answered all the reviewers’ questions and followed their suggestions by adding more descriptions and explanations.

(Q1) For the comment, “1) In vivo and in vitro in the article should be corrected as in vivo and in vitro.”,

(A1) The words “in vitro”, “in vivo” and “ex vivo” were changed into italic “in vitro”, “in vivo” and “ex vivo”.

(Q2) For the comment, “2) Page 8, line 290, [18F]17exhibited should be [18F]17 exhibited.”,

(A2) The words “[18F]17exhibited” were changed into “[18F]17 exhibited” in page 8.

(Q3) For the comment, “3) In the conclusion section, if the authors could forecast future directions of molecular imaging of hypoxia, it would be helpful to the readers.”,

(A3) The following paragraphs were added into the Conclusion and Perspectives section in page 28 to provide future directions of molecular imaging of hypoxia.

“…Moreover, in order to visualize hypoxia as molecular imaging, several study will be achieved for better molecular imaging in the future. First, proper radiotracers with more specific and highly selective to hypoxia should be developed for a variety of application to applied to clinical study. Thus, future study should focus on production of novel promising structures. Second, preparation process for radiotracers should be achieved via more simples and efficient steps. Particularly, short, low-cost, and environmentally synthetic process is useful. Third, radiolabeling protocol should also be more easily and effective, thus desired radiolabeled compounds should be obtained with high radiochemical yields. Fourth, improved imaging techniques and equipment should also be developed to visualize hypoxia more clearly. It is expected that many scientists will endeavor to synthesize and evaluate novel hypoxia radiotracers with better properties for clinical application in the future..…”.

We hope that our modifications to the manuscript for the specific concerns and questionable points will satisfy the reviewers and the requirements for the publication of this manuscript.

Round 2

Reviewer 2 Report

Thanks the author for the introduced corrections. I guess they will also agree that their review paper is bringing more clear  message to the readers except the extensive review on the avilable data.

I noticed only minor lexical errors and would like to draw authors attention to them:

in a couple of places I noticed "probable lypophilicity"/"probable pharmacokinetics"(line 915) may be they mean "favourable". And also,  "proper hydrophilic" lines 933 and 939, may be they mean proper hydrophilicity. Please, check in some other places the more appropriate use of the adverd or the adjective.

In conclusion, I'm satisfied by the corrections and recommend the publication in its current form. I also believe that it will be useful for the researcher working in the field.

I noticed only minor lexical errors and would like to draw authors attention to them:

in a couple of places I noticed "probable lypophilicity"/"probable pharmacokinetics"(line 915) may be they mean "favourable". And also,  "proper hydrophilic" lines 933 and 939, may be they mean proper hydrophilicity. Please, check in some other places the more appropriate use of the adverd or the adjective.